# Cancer exosomes induce tumor innervation

Marianna Madeo [1], Paul L. Colbert [1], Daniel W. Vermeer [1], Christopher T. Lucido [1], Jacob T. Cain [2], Elisabeth G. Vichaya [3], Aaron J. Grossberg [3,4], DesiRae Muirhead [5], Alex P. Rickel[6], Zhongkui Hong [6], Jing Zhao [7], Jill M. Weimer [2], William C. Spanos [1,8], John H. Lee[9], Robert Dantzer [3] & Paola D. Vermeer [1]

Patients with densely innervated tumors suffer with increased metastasis and decreased survival as compared to those with less innervated tumors. We hypothesize that in some tumors, nerves are acquired by a tumor-induced process, called axonogenesis. Here, we use PC12 cells as an in vitro neuronal model, human tumor samples and murine in vivo models to test this hypothesis. When appropriately stimulated, PC12 cells extend processes, called neurites. We show that patient tumors release vesicles, called exosomes, which induce PC12 neurite outgrowth. Using a cancer mouse model, we show that tumors compromised in exosome release are less innervated than controls. Moreover, in vivo pharmacological blockade of exosome release similarly attenuates tumor innervation. We characterize these nerves as sensory in nature and demonstrate that axonogenesis is potentiated by the exosome-packaged axonal guidance molecule, EphrinB1. These findings indicate that tumor released exosomes induce tumor innervation and exosomes containing EphrinB1 potentiate this activity.

[1] Cancer Biology and Immunotherapies Group, Sanford Research, 2301 East 60th St north, Sioux Falls, SD 57104, USA. [2] Pediatrics and Rare Diseases Group, Sanford Research, 2301 East 60th St north, Sioux Falls, SD 57104, USA. [3] Department of Symptom Research, MD Anderson Cancer Center, 1515 Holcombe Blvd, Unit 384, Houston, TX 77030, USA. [4] Department of Radiation Medicine, Cancer Early Detection Advanced Research Center, Oregon Health and Science University, 2720 SW Moody Ave KR-CEDR, Portland, OR 97201, USA. [5] Sanford Health Pathology Clinic, Sanford Health, 1305 West 18th St, Sioux Falls, SD 57105, USA. [6] Biomedical Engineering Program, University of South Dakota, 4800 North Career Ave, Sioux Falls, SD 57107, USA. [7] Population Health Group, Sanford Research, 2301 East 60th St north, Sioux Falls, SD 57104, USA. [8] Sanford Ears, Nose and Throat, 1310 West 22nd St, Sioux Falls, SD 57105, USA. [9] NantKwest, 9920 Jefferson Blvd, Culver City, CA 90232, USA. These authors contributed equally: Marianna Madeo, Paul L. Colbert. Correspondence and requests for materials should be addressed to P.D.V. (email: Paola.Vermeer@sanfordhealth.org)

Innervated tumors are more aggressive than less innervated ones[1–4]. For instance, in prostate cancer, recruitment of nerve fibers to cancer tissue is associated with higher tumor proliferative indices and a higher risk of recurrence and metastasis[2]. Denervation studies in mouse cancer models support a functional contribution of nerves in disease progression[5,6]. These studies strongly indicate that the nervous system is not a bystander but instead an active participant in carcinogenesis and cancer progression. However, a mechanistic understanding of how tumors obtain their neural elements remains unclear. Tumors may acquire innervation by growing within innervated tissues; in other words, nerves are already present within the microenvironment and the tumor acquires them by default. However, the clinical findings that some tumors of the same tissue are more innervated than others indicate instead an active, tumor-initiated process, similar to angiogenesis and lymphangiogenesis. The possibility that tumors invoke their own innervation, termed axonogenesis, has not been extensively explored[7,8].

Extracellular release of neurotrophic factors [e.g. nerve growth factor (NGF)] by tumor cells can contribute to cancer progression[9,10]. While such a mechanism likely contributes to tumor innervation, tumors release additional components that may also directly promote axonogenesis. Among these are extracellular vesicles such as exosomes. Exosomes are 30–150 nm vesicles that package a rich cargo (proteins, DNA, RNA, and lipids). Because they are generated by invagination of endocytic vesicles, the topology of exosomal transmembrane proteins is preserved as is, presumably, their biological activity. Exosomes are released into the extracellular milieu by most, if not all, cells[11] and function as vehicles of intercellular communication[12,13]. Mounting evidence supports the hypothesis that tumor-released exosomes promote disease progression through a number of mechanisms, including the induction/promotion of metastasis and tumor tolerance[13,14]. We hypothesized that one mechanism utilized by cancer cells to promote disease progression is the induction of tumor innervation. Here, we show that tumor released exosomes mediate axonogenesis in cancer and that this innervation is sensory in nature.

We utilize PC12 cells, a rat pheochromocytoma cell line, as an in vitro screen for axonogenic activity. When appropriately stimulated, PC12 cells extend neurites and morphologically resemble neurons. Our data with human samples indicate that exosomes from head and neck cancer patient tumors and matched blood induce significantly more neurite outgrowth from PC12 cells than exosomes from non-cancer control blood and tissue. To gain further mechanistic insight into this phenomenon, we turned to a murine model of human papillomavirus induced (HPV +) oropharyngeal squamous cell carcinoma (OPSCC). This model consists of oropharyngeal epithelial cells from C57Bl/6 mice that stably express HPV16 viral oncogenes, E6 and E7, H-Ras and luciferase (mEERL cells)[15,16]. We show that exosomes purified from the conditioned media of these cells induce neurite outgrowth of PC12 cells. When we compromise exosome release of mEERL cells by CRISPR/Cas9 genomic modulation of Rab27A/B, we find a significant attenuation of this activity. Moreover, mice implanted with the Rab27A/B modified tumors are sparsely innervated. As an additional test of this hypothesis, when mEERL tumor bearing mice are treated with the exosome release inhibitor, GW4869, tumor innervation is significantly reduced compared to that occurring in vehicle-treated mice.

Many molecules contained as cargo in exosomes are potential candidates for exosome-induced axonogenesis. We experimentally excluded NGF and focused on EphrinB1 as its increased expression leads to aggressive disease in different human cancers as well as in the mEERL mouse model[17–21]. EphrinB1 is a single pass transmembrane protein ligand that binds and activates the Eph receptor

tyrosine kinases; furthermore, EphrinB1 itself becomes phosphorylated and initiates its own signaling[22]. During development, EphrinB1 functions as an axonal guidance molecule[23]. Similar to other ephrins, EphrinB1 is frequently associated with growth cone collapse and inhibition of axon outgrowth[24–26], potentially redirecting axonal trajectory. We wondered if the contribution of EphrinB1 to disease progression extends beyond its signaling capabilities to include its axonal guidance properties. Here, we show that tumor released exosomes with a high EphrinB1 content potentiate neurite outgrowth of PC12 cells in vitro while compromised EphrinB1 expression or function significantly attenuates it. Consistent with these findings, mEERL tumors over-expressing EphrinB1 are significantly more innervated than those with basal EphrinB1 expression. Taken together, these data indicate that tumor released exosomes contribute to axonogenesis and that exosomal EphrinB1 potentiates this activity.

## Results

**Head and neck squamous cell carcinomas are innervated**. Immunohistochemical (IHC) staining of formalin-fixed paraffin embedded human head and neck squamous cell carcinomas (HNSCCs) for β-III tubulin, a neuron specific tubulin isoform, demonstrates these tumors are innervated; individual β-III tubulin positive fibers are coursing throughout the tumor tissue (Fig. 1a, "Nerve twigs"). These "nerve twigs" should not be confused with perineural invasion (PNI)[27,28]. PNI refers to tumor invading into nerves along the perineural space; axonogenesis refers to nerves invading into tumor. Within the perineural sheath, β-III tubulin positive fibers are tightly packed together in a very organized manner (Fig. 1a, "nerve bundle"). The β-III tubulin positive nerve fibers we identified are instead coursing as individual, unorganized twigs lacking a perineural sheath (Fig. 1a "nerve twigs"). Additional IHC staining shows that HNSCCs are negative for Tyrosine Hydroxylase (TH, sympathetic marker) and Vasoactive Intestinal Polypeptide (VIP, parasympathetic marker) but positive for Transient Receptor Potential Vanilloid type 1 channel (TRPV1, sensory marker) (Fig. 1b) indicating sensory innervation. Double IHC characterization of these neural "twigs" demonstrates co-localization of β-III tubulin (pink, arrows) and TRPV1 channels (brown, arrowheads) (Fig. 1c). Interestingly, some positively labeled fibers appeared to be nucleated, though their identity remains unclear. Regardless of their identity, we proceeded to quantify positively stained (unnucleated) fibers using stereology. We found that irrespective of whether tumors were densely or sparsely innervated, most of the β-III tubulin positive fibers were also immuno-positive for TRPV1 (Fig. 1d). On average, 83.7% ± 6.5 of β-III tubulin positive fibers were also TRPV1 positive. These data are consistent with sensory tumor innervation.

**Validation of exosome purification**. Prior to testing the contribution of exosomes to axonogenesis, we validated our differential ultracentrifugation exosome purification technique[29]. Exosomes were purified from human plasma; exosomes from human tissue samples were similarly purified from conditioned media collected after 48 h in culture. Scanning electron microscopic analysis of exosome preparations purified from normal donor plasma yielded vesicles consistent in shape and size (30–150 nm) with exosomes (Fig. 2a). Atomic force microscopy confirms a 63–105 nm size (Fig. 2b) and nanoparticle tracking analysis (for counting and sizing exosomes) (Fig. 2c) likewise indicates a size distribution consistent with exosomes[30]. These data indicate that our purification method yields vesicles consistent in size and shape with exosomes. We further characterized exosomes by western blot analysis for the exosome markers, CD9

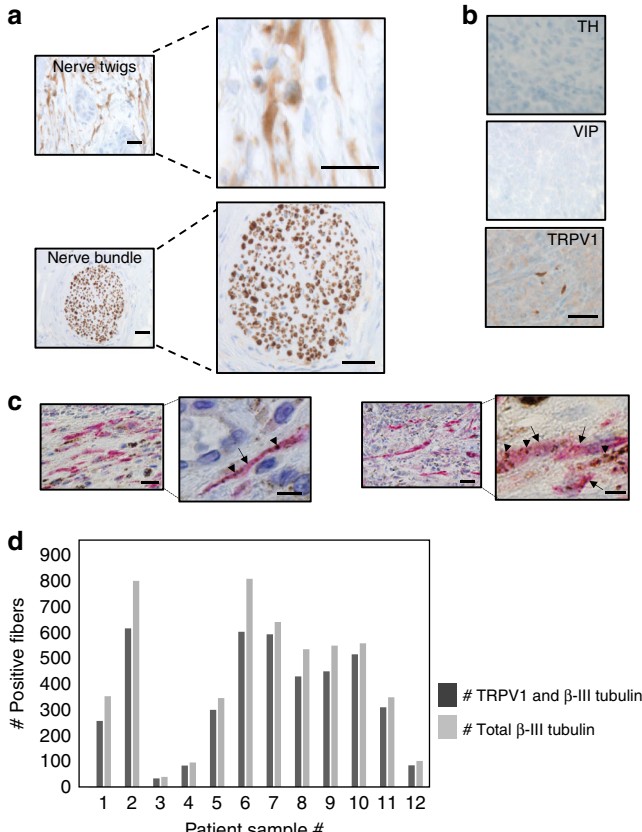

**Fig. 1** Sensory nerves innervate head and neck squamous cell carcinomas. **a** En face bright field images of HNSCC nerve "twigs" and "bundle" IHC stained for β-III tubulin (n = 30 patient tumors analyzed) (scale bars, 50 μm); brown, immunostain; blue, counterstain; **b** En face bright field images of HNSCC IHC stained for Tyrosine Hydroxylase (TH), Vasoactive Intestinal Polypeptide (VIP) and Transient Receptor Potential Vanilloid-type 1 channel (TRPV1) (scale bar, 20 μm) (n = 15 additional patient tumors analyzed for these markers). Brown, immunostain; blue, counterstain. Representative images shown. **c** En face bright field representative IHC images of human HNSCCs for β-III tubulin (pink, arrows) and TRPV1 (brown, arrowheads). Low magnification, scale bar, 50 μm. Higher magnification, scale bar, 20 μm. n = 12 patient tumors were co-stained with representative images presented. **d** Quantification of β-III tubulin/TRPV1 IHC stained human samples described in **c**. Dark gray bars, number of TRPV1 and β-III tubulin positive twigs; light gray bars, total number of β-III tubulin positive twigs

and CD81. We found that exosomes purified from both control and patient plasma were positive for one or both of these markers as were normal tonsil tissue and tumor released exosomes (Fig. 2d, Supplementary). The tonsil serves as control tissue since the majority of HPV + OPSCCs arise there.

**Cancer patient exosomes induce neurite outgrowth**. To test our hypothesis that tumor released exosomes induce axonogenesis, we utilized PC12 cells, a rat pheochromocytoma cell line, as an in vitro assay. When stimulated with NGF (100 ng/ml), PC12 cells differentiate into neuron-like cells extending neurites[31]. To test whether exosomes harbor neurite outgrowth activity, PC12 cells were stimulated with 3 μg of exosomes that were harvested from either 10 ml of blood or the conditioned media of the matched tumor tissue from eight head and neck cancer patients (Pt1–Pt8). We similarly collected blood from three healthy volunteers (Nl1-Nl3) and adult tonsils from two non-cancer

donors (TL1, 2). PC12 cells were fixed 48 h later and immunestained for β-III tubulin. Neurite outgrowth was quantified and the number of neurites compared. Given the time to procure these matched (blood and tumor) human samples, patient exosomes were not tested at the same time or on the same plate and thus are graphed separately (Fig. 3a, b). Consistent with the literature, we found that untreated PC12 cells extend few β-III tubulin positive neurites while those stimulated with NGF do so robustly. Exosomes from patients (both plasma and tumor) stimulated significant neurite outgrowth of PC12 cells while control plasma and tonsil exosomes had minimal neurite outgrowth activity (Fig. 3a, b). All comparisons can be found in Supplementary Tables 1–3. These data indicate that exosomes from head and neck cancer patients harbor neurite outgrowth activity that is absent in the exosomes of healthy controls. Western blot analysis of exosomes for EphrinB1 suggests that EphrinB1 is not required for neurite outgrowth activity in this assay (Supplementary Figure 1).

**mEERL tumors are innervated**. To determine whether the murine model of HNSCC we use is innervated similar to its human counterpart, mice were injected with mEERL cells in the hind limb ($1 \times 10^5$ cells/mouse), tumors harvested at endpoint, fixed, embedded and IHC stained for β-III tubulin, TH, VIP, and TRPV1. Similar to patient HNSCCs, mEERL tumors harbored β-III tubulin positive nerve twigs that were sensory in nature (TRPV1 positive) (Fig. 4a). Recent reports indicate that stress experienced by cancer cells (e.g. low oxygen) can induce their expression of β-III tubulin[32,33]. To address this possibility, serial sections of mEERL tumors were IHC stained for cytokeratin (epithelial marker) and β-III tubulin (neuronal marker) and found to display starkly different staining patterns (Fig. 4b), confirming that tumor cells and β-III tubulin positive fibers are distinct entities. To further characterize the nature of mEERL tumor innervation, we performed triple immunofluorescent labeling for β-III tubulin, TRPV1 and Tau (another neuronal marker) on formalin fixed paraffin embedded tumors. We found that β-III tubulin positive fibers were also positive for TRPV1 and Tau (merge, Fig. 4c). Taken together, these data are consistent with sensory innervation of mEERL tumors similar to the innervation in patient HNSCC tumors.

**mEERL exosomes induce neurite outgrowth**. Prior to testing neurite outgrowth activity of mEERL released exosomes, we characterized these exosomes as follows: exosomes purified from mEERL conditioned media were analyzed by electron microscopy (Supplementary Figure 2a), atomic force microscopy (Supplementary Figure 2b) and nanoparticle particle analysis (Supplementary Figure 2c) and, similar to their human counterparts, were consistent in size and shape with exosomes. Recent published studies suggest that more stringent methods are critical for eliminating other vesicles and cellular debris from exosome purifications[34]. One such method requires the addition of density gradient centrifugation[35]. To test whether this more stringent methodology purifies exosomes with neurite outgrowth activity, conditioned media from mEERL cells stably over-expressing EphrinB1 (mEERL EphrinB1) was collected, subjected to differential ultracentrifugation and subsequently to density gradient centrifugation. Fifteen fractions were collected and fractions 4–13 were analyzed by western blot for the exosomal markers CD9 and CD81. Consistent with the published literature, CD9+ and CD81 + vesicles were present in fraction 8;[36,37] exosomes purified by differential ultracentrifugation alone ("crude" sample) were also CD9+/CD81+ (Supplementary Figure 2d). To determine which fractions induce neurite outgrowth, fractions 4, 5, 8, 13, and

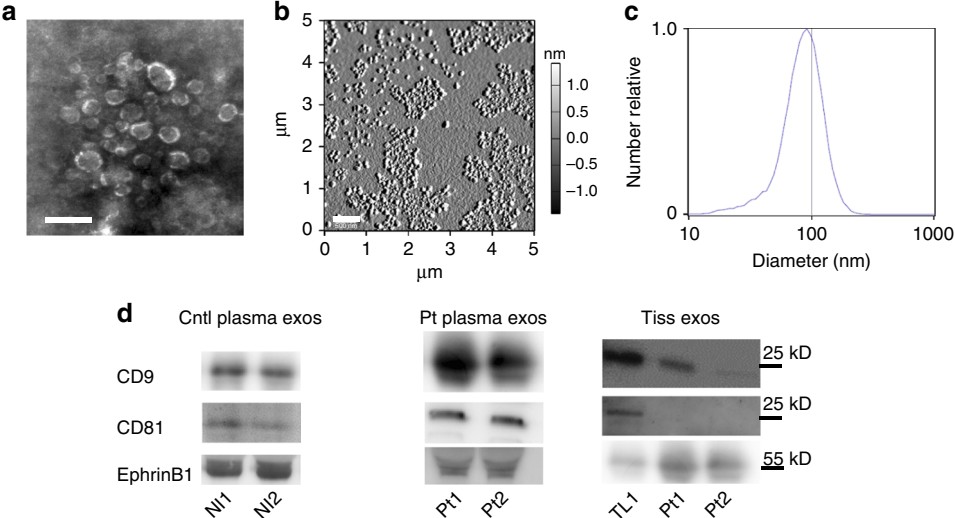

**Fig. 2** Validation of human exosome purification. **a** Scanning electron micrograph of exosomes purified from human plasma. Scale bar, 200 nm. $n = 4$ biological samples. **b** Atomic force microscopy amplitude trace of exosomes purified from human plasma. Size range, 63–105 nm. Scale bar, 500 nm. $n = 7$ biological samples. **c** Representative nanoparticle tracking analysis of exosomes purified from human plasma. $N = 4$ biological samples. **d** Western blot analysis of control (Cntl) and patient (Pt) exosomes (exos). Western blots have been cropped for clarity and conciseness. Nl1-2 plasma from healthy volunteers, TL 1 normal adult tonsil. $n = 3$ technical replicates

"crude" exosomes were tested on PC12 cells. While fraction 8 and "crude" exosomes demonstrate neurite outgrowth activity, CD9 negative fractions 4, 5, and 13 lacked this activity (Supplementary Figure 2e; all comparisons found in Supplementary Table 7). These data indicate that inclusion of density gradient centrifugation concentrates CD9+/CD81+ exosomes to a single fraction that retains neurite outgrowth activity. Interestingly, CD9+/CD81+ fractions are also positive for EphrinB1 (Supplementary Figure 2d) indicating that neurite outgrowth activity is retained in EphrinB1 positive CD9+/CD81+ exosomes.

**Inhibition of exosome release attenuates innervation**. To test the hypothesis that tumor released exosomes induce tumor innervation, we utilized CRISPR/Cas9 to genetically modify *Rab27A* and *Rab27B* in mEERL parental cells. These two small GTPases contribute to exosome release and their knock-down compromises release of CD9+ exosomes[38,39]. The clone generated is heterozygous for *Rab27A* and homozygous deleted for *Rab27B* (mEERL *Rab27A*$^{-/+}$ *Rab27B*$^{-/-}$). For characterization and testing of this clone, exosome samples were normalized to producing cell number. Exosomes isolated from *Rab27A*$^{-/+}$ *Rab27B*$^{-/-}$ cells displayed less CD9 signal than parental mEERL exosome samples; this is consistent with previous studies demonstrating a decreased capacity to release CD9 positive exosomes by cells compromised in Rab27A/B expression (Supplementary Figure 3a)[40]. Exosomes were further analyzed by nanoparticle tracking analysis (Supplementary Figure 3b) and membrane permeant fluorescein labeling (CFDA-SE) (Supplementary Figure 3c) indicating decreased exosome release by the *Rab27A*$^{-/+}$ *Rab27B*$^{-/-}$ mutant cells.

To test whether compromised exosome release affects neurite outgrowth of PC12 cells, mEERL parental and mEERL *Rab27A*$^{-/+}$ *Rab27B*$^{-/-}$ cells were cultured in vitro, exosomes purified from conditioned media, normalized to producing cell number and equivalent volumes applied to PC12 cells ($n = 4$ wells/condition; experiment repeated at least 3 times). The neurite outgrowth activity of mEERL *Rab27A*$^{-/+}$ *Rab27B*$^{-/-}$ exosomes was significantly attenuated compared to that of exosomes from mEERL parental cells (Fig. 5a; all comparisons found in Supplementary Table 4). To test whether compromised exosome

release alters innervation in vivo, C57Bl/6 mice ($n = 7$ mice/group) were implanted with mEERL parental or *Rab27A*$^{-/+}$*Rab27B*$^{-/-}$ cells; tumors were harvested and 15 μg of whole tumor lysate quantified by western blot for β-III tubulin, TRPV1 and Tau (Fig. 5b). Consistent with our hypothesis, *Rab27A*$^{-/+}$ *Rab27B*$^{-/-}$ tumors were significantly decreased in β-III tubulin, TRPV1 and Tau as compared to mEERL parental tumors (Fig. 5c–e). These data support our hypothesis that tumor released exosomes contribute to tumor innervation. Notably, *Rab27A*$^{-/+}$ *Rab27B*$^{-/-}$ tumors also grew significantly slower than mEERL parental tumors (Supplementary Figure 3d). Importantly, in vitro proliferation assays show that cell doubling time was not significantly different between mEERL parental and *Rab27A*$^{-/+}$ *Rab27B*$^{-/-}$ cells (Supplementary Figure 3e).

To further test whether exosomes contribute to tumor innervation, C57Bl/6 mice were implanted with mEERL parental tumors ($n = 7$ mice/group) and segregated into two groups. One day post tumor implantation, exosome release was systemically blocked by intraperitoneal injection with the cell permeable, neutral sphingomyelinase inhibitor GW4869 (1.25 mg/kg/day). This dose is based on previously published in vivo studies[41]. GW4869 blocks the ceramide-mediated secondary invagination of endosomes that generates the multi-vesicular body (MVB) from which exosomes are released. In this way, GW4869 inhibits release of mature exosomes[42]. Mice were treated six days a week, and administered a double dose (2.5 mg/kg) on the 6th day. Control animals were injected with vehicle. This schedule of injection continued until sacrifice criteria were met. Blood was collected from all mice at sacrifice (via cardiac puncture), exosomes purified by differential ultracentrifugation and quantified by nanoparticle tracking analysis. We found a significant reduction in exosomes from GW4869 treated animals indicating the drug successfully attenuated exosome release (Fig. 5f). To determine if this decreased exosome release affected tumor innervation, tumors ($n = 7$ tumors/group) were harvested at end point, embedded in paraffin and analyzed by IHC for β-III tubulin. Stained sections were scanned on an Aperio slide scanner, images extracted and analyzed using ImageJ software. We found significantly reduced β-III tubulin positive fibers in tumors of GW4869-treated mice as compared to tumors of

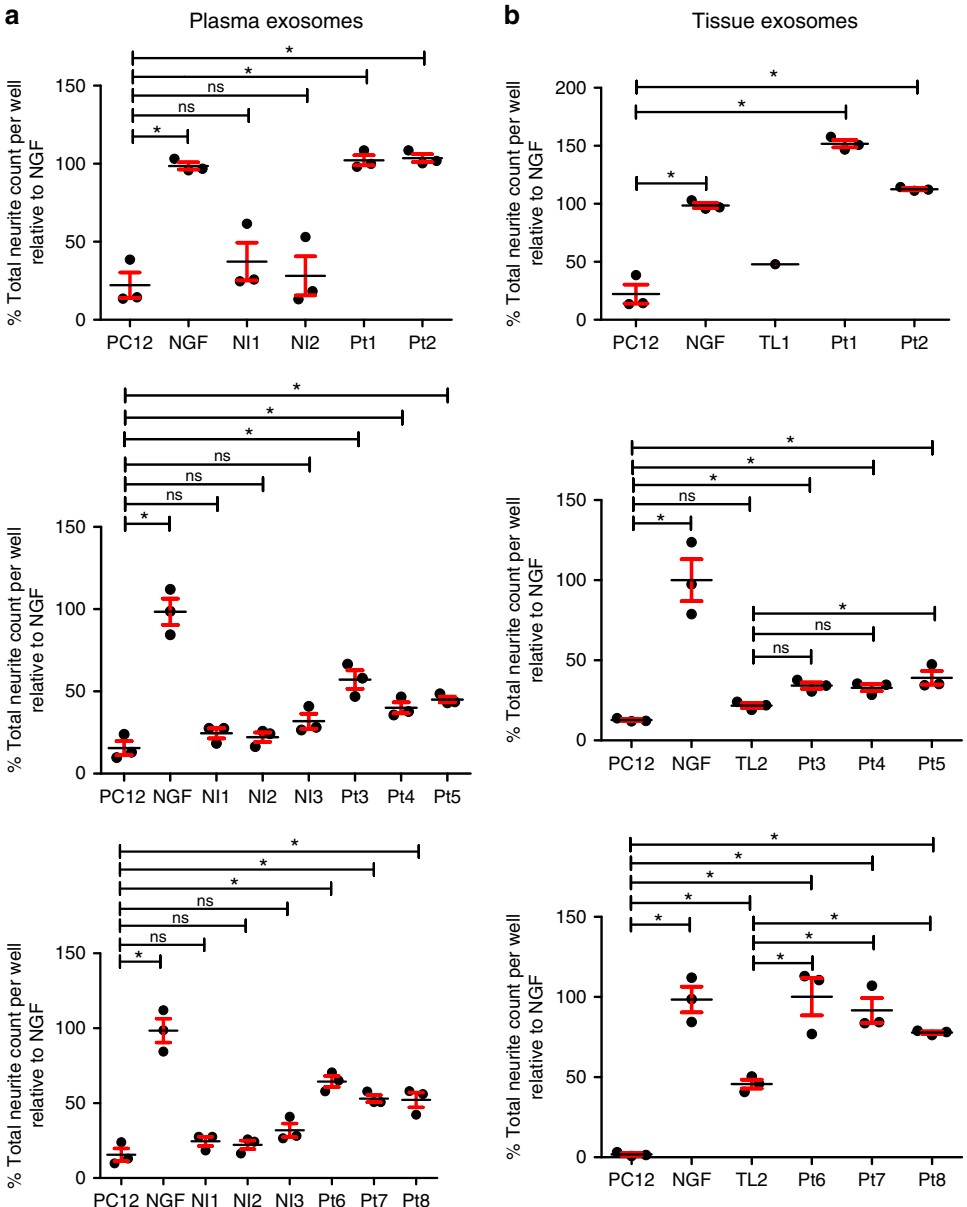

**Fig. 3** Head and neck cancer patient derived exosomes induce neurite outgrowth. Quantified neurite outgrowth of PC12 cells following plasma (**a**) or tissue (**b**) exosome stimulation. NGF (100 ng/ml) serves as a positive control. NI1- NI3 non-cancer control blood, Pt1-Pt8 cancer patients, TL1,2 normal tonsils. *n* = 3 technical replicates/sample; assay repeated at least *n* = 2 times with similar results. The exosome yield from TL1 was low and insufficient for multiple replicates. For statistical analysis, one-way ANOVA with post-hoc Fisher's Least Significant Difference (LSD) test was used; LSD *p* values reported; *$p \leq$ 0.03; ns, not significant, center value used was the mean. Error bars, standard deviation. The variance between groups compared is similar. All comparisons and LSD *p* values found in Supplementary Tables 1-3

vehicle-treated control mice (Fig. 5g). Representative photomicrographs of IHC stained tumor tissue for β-III tubulin show single fibers similar to those identified in human and mEERL tumors (Figs 5h, 4a, 1a).

**Exosome-mediated neurite outgrowth does not require NGF.** Previous reports indicate that neurotrophic factors secreted by tumor cells can induce tumor innervation[9,10]. mEERL cells express NGF but not BDNF, NT-3, NT-4/5 or GDNF (Geo Accession GSE68935)[43]. Thus, we wondered whether release of NGF by mEERL cells contributes to neurite outgrowth activity. To test this, conditioned media from mEERL parental and *EphrinB1* cells was incubated with neutralizing anti-NGF

antibody for 24 h and tested on PC12 cells. While NGF neutralization robustly attenuated neurite outgrowth of recombinant NGF, it had no effect on neurite outgrowth stimulated by conditioned media from either mEERL parental or *EphrinB1* cells indicating that soluble NGF is not required for neurite outgrowth activity in this assay (Fig. 6a; all comparisons found in Supplementary Table 4). It remains possible that NGF is packaged as cargo within exosomes and mediates neurite outgrowth in this way. To test this, whole cell lysates and purified exosomes from mEERL parental and *EphrinB1* cells were analyzed by western blot for NGF. We confirmed that mEERL parental and *EphrinB1* over-expressing cells produce NGF, as indicated by the presence of two bands corresponding to the processed forms of NGF (13.5 and 16.5 kD) in the whole cell lysate (WCL)[44]. These bands are

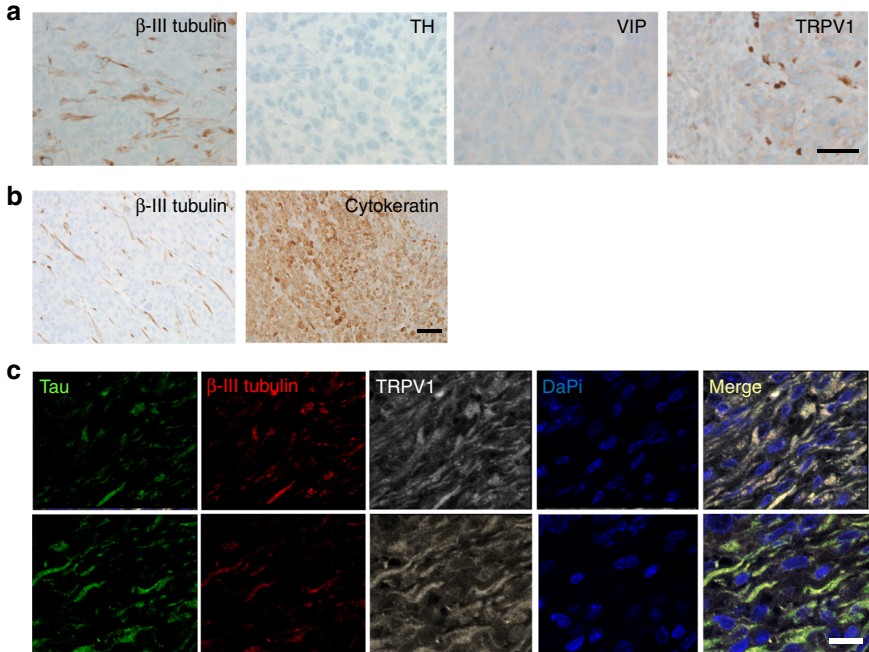

**Fig. 4** Sensory innervation of mEERL tumors. **a** IHC staining of mEERL tumors for β-III tubulin ($n = 30$); Tyrosine Hydroxylase (TH), Vasoactive Intestinal Polypeptide (VIP), and Transient Receptor Potential Vanilloid-type one (TRPV1) ($n = 15$ additional tumors analyzed for each marker) (scale bar, 20 μm). Brown, immunostain; blue, counterstain. **b** IHC stained serial sections of a mEERL parental tumor for β-III tubulin (neuronal marker) and cytokeratin (epithelial marker), scale bar, 50 μm. **c** Triple immunofluorescent labeling of mEERL tumor for Tau (green), β-III tubulin (red), and TRPV1 (far red, seen as white); merged image presented (merge). Nuclei counterstained with DaPi (blue). Scale bar, 10 μm

absent from CD9 + exosomes (Fig. 6b) indicating that NGF is not packaged within exosomes or required for exosome-mediated neurite outgrowth activity.

**EphrinB1 potentiates exosome neurite outgrowth activity.** We wondered what exosomal cargo was necessary and sufficient for neurite outgrowth activity. During development, EphrinB1 functions as an axonal guidance molecule[23]. Moreover, EphrinB1 has been identified as exosome cargo in other systems (Exocarta.org)[45] and significantly contributes to disease progression in HNSCC[46,17]. Thus, to test its contribution in axonogenesis, we generated EphrinB1 modified mEERL cell lines. Stable over-expression of wild-type EphrinB1 is referred to as mEERL EphrinB1[17]. CRISPR/Cas9 engineered mEERL cell lines compromised in EphrinB1 function or expression are denoted as follows: mEERL EphrinB1 Null1, or Null2 (two independent clones in which EphrinB1 is deleted) and extracellularly truncated EphrinB1 cells are denoted mEERL EphrinB1ΔECD. The characterization of these CRISPR lines is presented in Supplementary Figures 4–5.

To determine the contribution EphrinB1 to neurite outgrowth, exosomes were purified from the conditioned media of the various mEERL cell lines and tested on PC12 cells. Exosomes from mEERL parental cells significantly induced neurite outgrowth of PC12 cells while over-expression of EphrinB1 potentiated this activity (Fig. 6c; all comparisons found in Supplementary Table 5). MEERL EphrinB1ΔECD, Null1 and Null2 exosomes retained neurite outgrowth activity indicating that mEERL released exosomes promote neurite outgrowth and that EphrinB1 is not required for this activity, but significantly potentiates it.

Given that exosomes purified from mEERL EphrinB1 cells potentiated neurite outgrowth of PC12 cells, we tested whether it is packaged as exosome cargo. Consistent with the published

literature, EphrinB1 was indeed packaged within exosomes as assessed by western blot analysis (Fig. 6d)[45] (Exocarta.org). Moreover, while the extracellular domain of EphrinB1 was absent in mEERL EphrinB1ΔECD exosomes, the intracellular domain remained as cargo running at a similar molecular weight as (endogenously) proteolytically processed EphrinB1[47] (Fig. 6d).

The data indicate that exosomal EphrinB1 is not required for neurite outgrowth activity, but does potentiate it. Thus, we wondered if exosomal EphrinB1 activates known signaling pathways in PC12 cells that are important for neuritogenesis. One such pathway is the MAP Kinase pathway[48]. To test this, PC12 cells were stimulated for 5 minutes with either PBS (negative control), NGF (50 ng/ml, positive control), or exosomes purified from either mEERL parental, EphrinB1 or Null2 conditioned media. PC12 cells were then harvested and whole cell lysate analyzed by western blot. Stimulation with NGF induced phosphorylation of Erk1/2 while control (PBS) did not activate the MAP Kinase pathway. Interestingly, stimulation with mEERL parental, EphrinB1 and Null2 exosomes also induced a similar activation of this signaling pathway (Fig. 6e). These data suggest that binding of EphrinB1 to PC12 expressed receptor(s) is not required for initiation of MAP kinase signaling. The PC12 neurite outgrowth data with exosomes purified from these cell lines (Fig. 6c) indicate that exosomes from mEERL EphrinB1 cells potentiate neurite outgrowth of PC12 cells. It remains possible that EphrinB1 exosomes promote sustained MAP kinase signaling in PC12 cells and that this prolonged signaling results in potentiation of neurite outgrowth. Such a mechanism would be consistent with the published literature[49–52].

**High-risk HPV E6 and neurite outgrowth.** We have thus far tested neurite outgrowth activity from mEERL cells or their derivatives, all of which are HPV positive. We wondered whether exosome mediated neurite outgrowth activity extended to

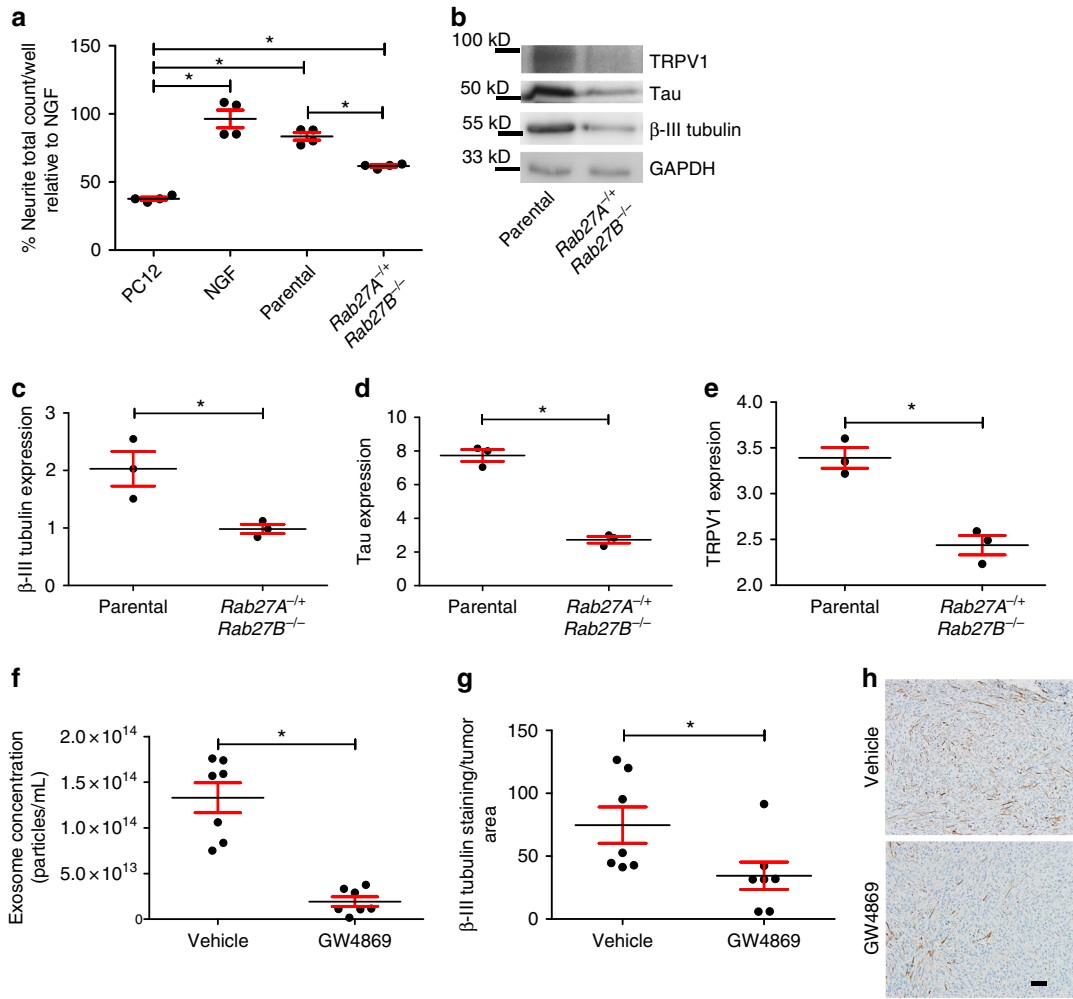

**Fig. 5** Compromised exosome release leads to sparsely innervated tumors. **a** Neurite outgrowth of PC12 cells following exosome stimulation from the indicated sources. Stimulation with recombinant NGF (100 ng/ml) serves as a positive control. $n = 4$ technical replicates/condition; experiment repeated at least three times with biological replicates. Statistical analysis by one-way ANOVA with post hoc Fisher's Least Significant Difference (LSD) test. LSD $p$ values reported; *$p \leq 0.02$. The variance between groups compared is similar. Central value used is the mean. All comparisons and $p$ values found in Supplementary Table 4. **b** Western blot analysis of whole tumor lysate from mice bearing mEERL parental or $Rab27A^{-/+}Rab27B^{-/-}$ tumors. $n = 3$ technical replicates. $n = 4$ biological replicates. Western blots have been cropped for clarity and conciseness. Western blot quantification by densitometry for **c** β-III tubulin, *$p<0.05$; **d** Tau, *$p<0.0001$; and **e** TRPV1, *$p<0.001$. Statistical analysis by unpaired, two-tailed Student's $t$-test. Central value used was the mean. The variance between groups compared is similar. **f** Nanoparticle tracking analysis of exosomes purified from the plasma of tumor bearing mice treated with GW4869 or vehicle. Statistical analysis by two-tailed Student's $t$-test. Central value used was the mean. The variance between groups compared is similar; *$p<0.0001$. **g** Quantification of β-III tubulin IHC staining of tumors from mice ($n = 7$/group) treated with GW4869 or vehicle; Statistical analysis by unpaired, two-tailed Student's $t$-test; *$p<0.05$. Central value used was the mean. Experiment performed once. The variance between groups compared is similar. **h** En face bright field representative images of IHC for β-III tubulin of mEERL tumors treated with GW4869 or vehicle. $n = 7$/group. Scale bar, 100 μm. All error bars are standard deviation

exosomes from non-HPV transformed cells. To test this, exosomes from two HPV negative human squamous cell carcinoma cell lines (UM-SCC1 and UM-SCC19, referred to as SCC1 and SCC19) and those from two HPV positive cell lines (UM-SCC47 and 93-VU-147T-UP-6, referred to as SCC47 and 147T) were tested on PC12 cells. Interestingly, though HPV negative exosomes harbored significantly less neurite outgrowth activity than HPV positive exosomes, they still induced neurite outgrowth from PC12 cells suggesting that HPV infection is not required for neurite outgrowth activity of exosomes (Fig. 7a; all comparisons found in Supplementary Table 5).

Since our data indicate that EphrinB1 potentiates neurite outgrowth activity of exosomes, we wondered if merely over-expressing EphrinB1 in an HPV negative squamous cell carcinoma cell line would increase neurite outgrowth activity of

its exosomes. Thus, we stably over-expressed EphrinB1 in SCC1 cells and tested their exosomes on PC12 cells. Importantly, SCC1 cells express endogenous EphrinB1 albeit at a low level (Supplementary Figure 7c). Exosomes from SCC1-EphrinB1 cells induced significantly more neurite outgrowth than those from the SCC1 parental cells (Fig. 7b; all comparisons found in Supplementary Table 6). These data suggest that HPV negative head and neck cancer patients that harbor elevated EphrinB1 expression may also harbor elevated tumor innervation potential. To test this concept in vivo, immune incompetent NOD SCID mice were implanted with either SCC1 or SCC1-EphrinB1 cells ($n = 4$ mice/group). Tumor innervation was assessed 10 days later. Whole tumor lysates ($n = 4$ mice/group) were analyzed by western blot (Fig. 7c). SCC-1 EphrinB1 tumors had significantly more β-III tubulin and Tau by western blot than SCC1 parental

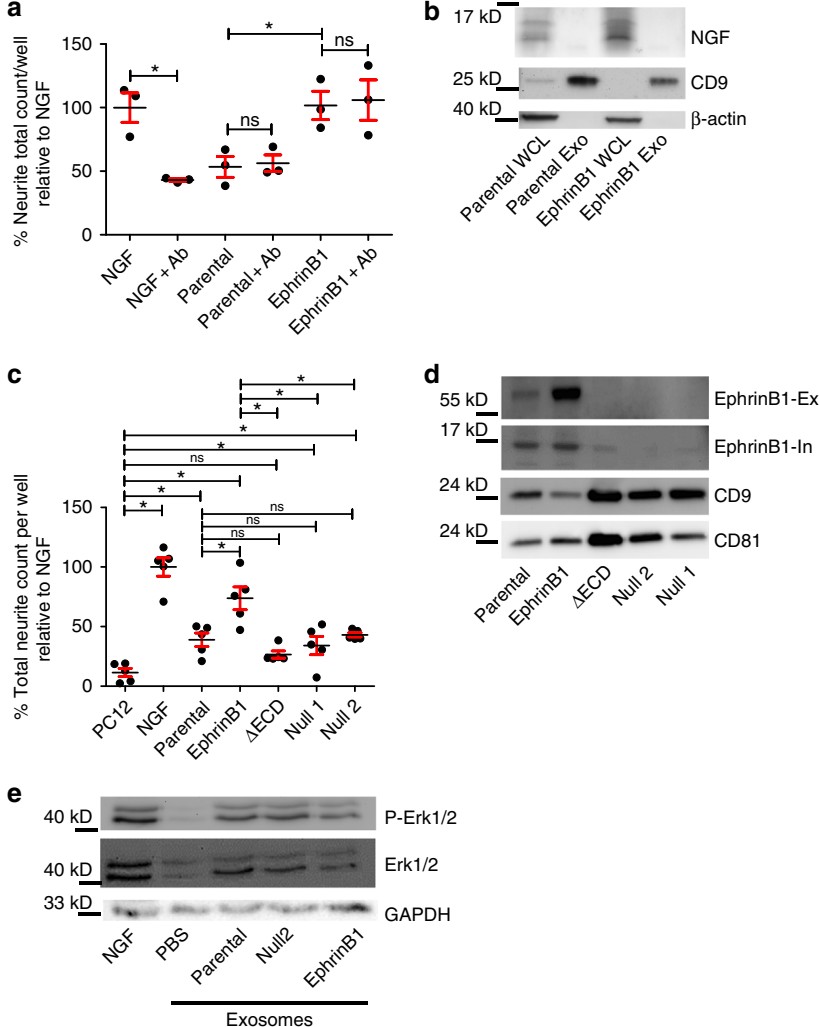

**Fig. 6** mEERL exosomes induce neurite outgrowth without NGF. **a** PC12 cells were stimulated with 100 ng/ml of recombinant NGF (NGF), mEERL parental (parental) or mEERL *EphrinB1* (EphrinB1) conditioned media with or without neutralizing anti-NGF antibody (+Ab); 24 h later, cells were fixed and stained for β-III tubulin expression which was then quantified. Statistical analysis by two-way ANOVA with post hoc Fisher's Least Significant Difference (LSD) test. LSD *p* values reported; *$p<0.006$. ns, not significant. The variance between groups compared is similar. $n = 3$ technical replicates/condition; experiment repeated at least two times with biological replicates. Central value used was the mean. All comparisons and LSD *p* values found in Supplementary Table 4. **b** Western blot analysis of exosomes (Exo) or whole cell lysate (WCL) from the indicated mEERL cell lines. Western blot repeated at least $n = 2$ times with biological replicates. **c** Neurite outgrowth quantification of PC12 cells following exosome treatment from the indicated sources. Statistical analysis by one-way ANOVA with post hoc Fisher's Least Significant Difference (LSD) test. LSD *p* values reported; *$p<0.02$; ns, not significant. $n = 5$ technical replicates/condition; experiment repeated at least twice with biological replicates. Central value used was the mean. The variance between groups compared is similar. All comparisons and LSD *p* values found in Supplementary Table 5. **d** Western blot analysis of exosomes purified from the indicated mEERL cell lines. EphrinB1-Ex, EphrinB1 extracellular epitope antibody. EphrinB1-In, EphrinB1 intracellular epitope antibody. Western blot repeated at least $n = 2$ times with biological replicates. **e** Western blot analysis of PC12 lysate following stimulation for 5 min with PBS, NGF (50 µg/ml) or exosomes purified from mEERL parental, *EphrinB1*, or *Null2* conditioned media. Experiment repeated at least three times with biological replicates. All error bars are standard deviation. Western blots have been cropped for clarity and conciseness

tumors (Fig. 7d, e) suggesting increased tumor innervation; tumor expression of TRPV1 was not significantly different (Fig. 7f). Although not directly relevant to the main theme of the present study, over-expression of EphrinB1 significantly enhanced tumor growth in vivo[17] (Supplementary Figure 7a). Individual tumor growth curves are presented in Supplementary Figure 7b. Taken together, these data indicate that exosomes from both HPV positive and HPV negative squamous cell carcinoma cell lines induce tumor innervation. Consistent with our other data, EphrinB1 potentiates this activity.

We were intrigued by the apparent contribution of HPV infection on exosome-mediated neurite outgrowth activity (Fig. 7a). Since E6 and E7 are the main viral oncogenes, we

focused on them. HPV16 induces OPSCC and is thus considered a high risk HPV;[53] low risk HPVs rarely cause cancer. One important difference between high and low risk HPVs is found in their E6 proteins; only high risk E6 contains a C-terminal PDZ binding motif (PDZBM) which contributes to oncogenic transformation[54,55]. In fact, E6's PDZBM mediates interactions with a number of different proteins that contribute to oncogenic activity[56–61]. To test if these interactions also contribute to exosome mediated neurite outgrowth activity, we tested exosomes from primary human tonsil epithelia (HTE), exosomes from cells stably expressing HPV16 E6 and E7 (HTE E6E7) and exosomes from cells in which the E6 PDZBM is deleted (HTE E6ΔE7). Expression of full length E6 and E7 was sufficient to induce

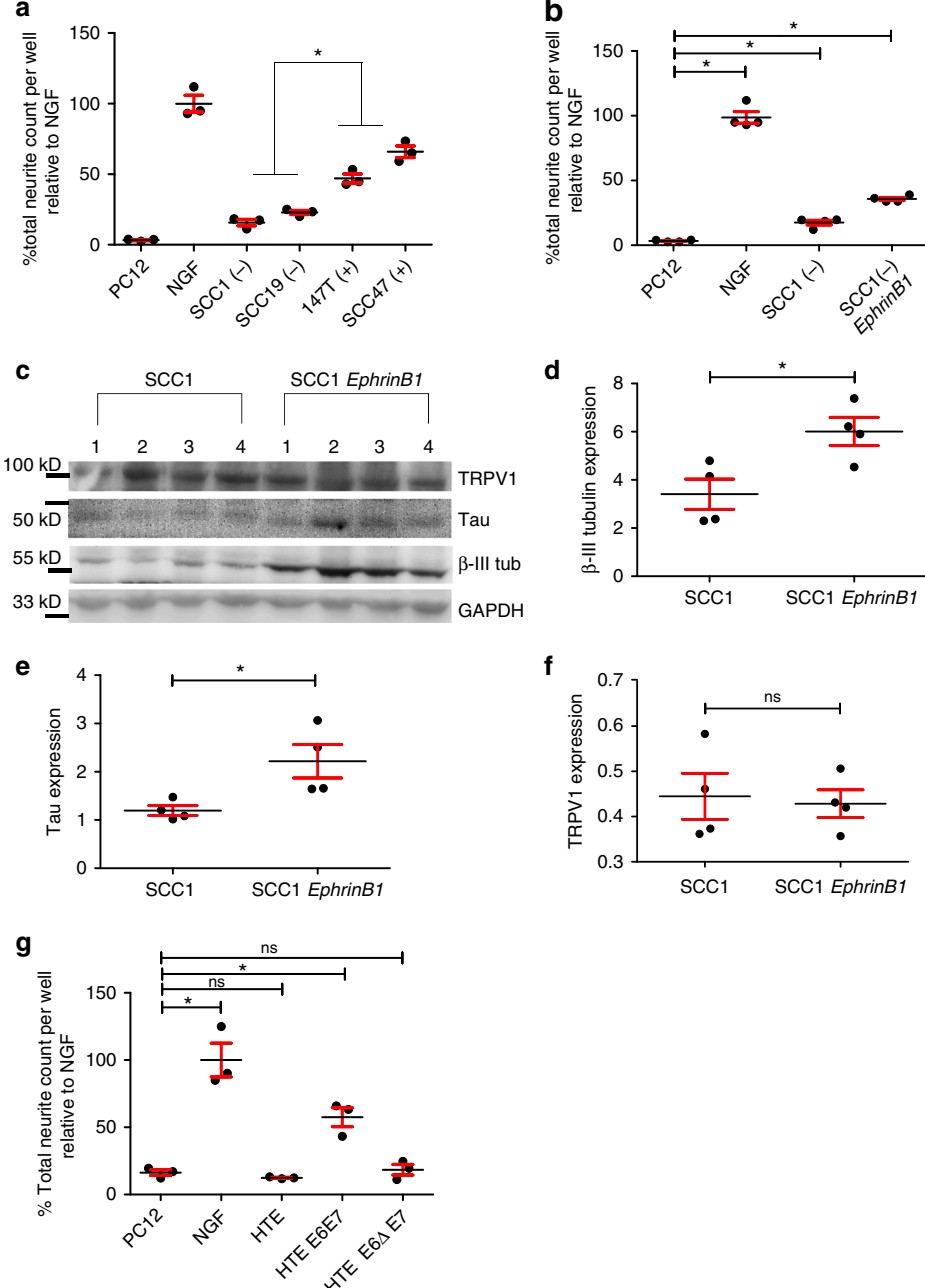

**Fig. 7** HPV status and EphrinB1 contribute to neurite outgrowth. **a** Quantified neurite outgrowth of PC12 cells following exosome stimulation with HPV negative (−) or positive (+) human cell lines; statistical analysis by one-way ANOVA with post hoc Tukey test for differences between HPV- and + groups; *$p<0.001$. The variance between groups compared is similar. $n = 3$ technical replicates/condition; experiment repeated at least two times. Central value was the mean. Comparisons between individual groups were analyzed by one-way ANOVA with post hoc Fisher's Least Significant Difference (LSD) test. All comparisons and LSD $p$ values found in Supplementary Table 5. **b** Quantified neurite outgrowth of PC12 cells following exosome stimulation from HPV(−) SCC1 parental cells or SCC1(−) EphrinB1 over-expressing cells [SCC1(−) EphrinB1]. Statistical analysis by one-way ANOVA with post hoc Fisher's Least Significant Difference (LSD) test; LSD $p$ values reported, *$p<0.02$. The variance between groups compared is similar. $n = 4$ technical replicates/condition; experiment repeated twice. Central value used was the mean. All comparison and LSD $p$ values found in Supplementary Table 6. **c** Western blot analysis of whole tumor lysates from mice bearing SCC1 or SCC1 EphrinB1 tumors. Western blots repeated at least $n = 4$ times with technical replicates. Western blots have been cropped for clarity and conciseness. Densitometric quantification of western blot in panel **c** for **d** β-III tubulin, **e** Tau, and **f** TRPV1. Statistical analysis by unpaired, two-tailed Student's $t$-test; *$p<0.03$; ns not significant. The variance between groups compared is similar. Central value was the mean. **g** Quantified neurite outgrowth of PC12 cells following stimulation with exosomes from HTE, human tonsil epithelia; HTE E6ΔE7, cells expressing HPV16 E6 deleted of its PDZBM(Δ) and E7; HTE E6E7, cells expressing HPV16 E6 and E7. Statistical analysis by one-way ANOVA with post hoc Fisher's Least Significant Difference (LSD) test, LSD $p$ values reported, *$p<0.0001$; ns, not significant. The variance between groups compared is similar. Central value was the mean. $n = 3$ replicates/condition; experiment repeated twice. All comparisons and LSD $p$ values found in Supplementary Table 6. All error bars are standard deviation

neurite outgrowth activity while deletion of E6 PDZBM abrogated this effect (Fig. 7g; all comparisons found in Supplementary Table 6). These data suggest that HPV16 E6 PDZBM modulates exosome content or quality thereby affecting neurite outgrowth activity. In addition and consistent with our other findings, we found that these human cell lines express EphrinB1 to varying degrees and that expression did not appear to correlate with PC12 neuritogenesis (Supplementary Figure 7c). These data are consistent with our findings suggesting that EphrinB1 is not required for exosome-mediated neurite outgrowth activity.

To gain further insights on the contribution of EphrinB1 to tumor innervation, C57Bl/6 mice were implanted with either mEERL parental, *EphrinB1*, *ΔECD*, *Null1*, or *Null2* cells ($n = 10$ mice/group). Fourteen days post-implantation, animals were sacrificed and tumors harvested; $n = 5$ tumors/group were utilized for whole tumor lysate and western blot analysis for β-III tubulin, Tau and TRPV1. Signals were normalized to GAPDH and quantified by densitometry. We found that mEERL *EphrinB1* tumors harbored significantly more β-III tubulin, TRPV1 and Tau positive fibers as compared to mEERL parental, *Null1* and *Null2* tumors (Supplementary Figure 8a–d; all comparisons found in Supplementary Tables 6–8), consistent with our other data indicating that EphrinB1 potentiates tumor innervation. Interesingly, the extent of innervation (β-III tubulin and Tau) was similar between mEERL *EphrinB1* and mEERL *EphrinB1ΔECD* tumors (Supplementary Figure 8b, c), however mEERL *EphrinB1ΔECD* tumors possessed significantly less TRPV1 expression (Supplementary Figure 8d). These data indicate that while mEERL *EphrinB1ΔECD* tumors are as highly innervated as mEERL *EphrinB1* tumors (having the same extent of β-III tubulin and Tau expression), this increased innervation is not all TRPV1 positive. Thus, the data suggest that the type of innervation may be different in the mEERL *EphrinB1ΔECD* tumors as compared to the mEERL *EphrinB1* tumors. Tumor growth curves are presented in Supplementary Figure 8e. Consistent with the published data, over-expression of EphrinB1 results in increased tumor growth[17]. We tested if these neural proteins were expressed by mEERL cells themselves via western blot analysis of whole cell lysates. Supplementary Figure 8f show that while whole tumor lysate (mEERL *EphrinB1*) was positive for β-III tubulin, Tau and TRPV1, whole cell lysates from mEERL parental cells and its derivative cell lines were negative for these neuronal markers indicating that the cancer cells obtain nerves in vivo.

**Exosomes from other cancers induce PC12 neurite outgrowth.** While the laboratory focus is HPV-induced HNSCC, we wondered whether our findings extended to other types of cancers. Thus, we harvested exosomes from conditioned media of CT26 (colorectal cancer cell line), B16 (melanoma cell line) and 4T1 (breast cancer cell line) cells, verified they expressed CD9 (Fig. 8a) and tested their ability to induce neurite outgrowth of PC12 cells. Similar to human and mouse HNSCC, exosomes from these murine cancer cell lines induced significant neurite outgrowth of PC12 cells (Fig. 8b; all comparisons found in Supplementary Table 6). Consistent with our findings with mEERL and human exosomes, EphrinB1 was not required for exosome-induced neurite outgrowth as only B16 exosomes packaged EphrinB1 (Supplementary Figure 9a). Interestingly, B16 released exosomes also demonstrated the most robust neurite outgrowth activity (Fig. 8b). These data extend our findings and suggest that tumor released exosomes may mediate tumor innervation in a wide variety of solid tumors though this hypothesis requires extensive testing.

## Discussion
Our findings propose a new mechanism for tumor-induced, exosome-mediated axonogenesis. We show that human and mouse HPV + HNSCCs are innervated de novo by TRPV1, Tau, β-III tubulin positive sensory nerves. Moreover, while mEERL tumors secrete NGF, NGF is not required for neurite outgrowth activity in our in vitro assay nor is it packaged within exosomes. Mechanistically, packaging of full length EphrinB1 as exosome cargo significantly potentiates neurite outgrowth in vitro and tumor innervation in vivo and its deletion reverses the EphrinB1 mediated potentiation. While the PC12 assay would have predicted that mEERL *EphrinB1ΔECD* tumors to be sparsely innervated in vivo, we instead found that they were as innervated (based on β-III tubulin and Tau) as mEERL *EphrinB1* tumors (Supplementary Figure 8a–d). These data emphasize the complexity of the tumor microenvironment that cannot be replicated in the PC12 in vitro assay as exosomes have the potential to act on multiple cell targets to induce axonogenesis. Despite this, both systems indicate that EphrinB1 potentiates tumor innervation and, perhaps more importantly, that full-length EphrinB1 is not required for tumor innervation. In addition, we show that compromising release of CD9 + exosomes results in significantly decreased innervation in vivo, further implicating tumor released exosomes as mediating tumor innervation. These pre-clinical studies are supported by findings with human HNSCC samples

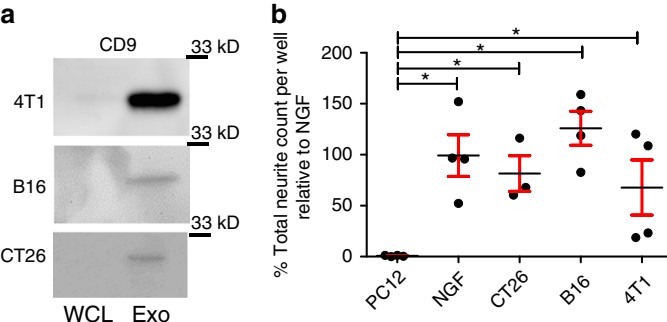

**Fig. 8** Exosomes from multiple cancer cell lines harbor neurite outgrowth activity. **a** Western blot analysis of whole cell lysate (WCL) or exosomes (Exo) from the indicated cell lines. Western blots have been cropped for clarity and conciseness. **b** Quantification of neurite outgrowth of PC12 cells following exosome stimulation from the indicated cell lines. Statistical analysis by one-way ANOVA with post hoc Fisher's Least Significant Difference (LSD) test. LSD $p$ values reported; *$p<0.01$. All comparisons and LSD $p$ values found in Supplementary Table 6. The variance between the groups statistically compared is similar. Central value used was the mean. $n = 4$ technical replicates/condition; experiment repeated at least 3 times with biological replicates. Error bars, standard deviation

where matched HNSCC patient plasma and tumor exosomes harbor neurite outgrowth activity. Taken together, these data indicate that CD9 + exosomes released by HPV-positive tumor cells promote tumor innervation in vivo. We also provide in vitro evidence that exosomes from other cancer cell lines also harbor neurite outgrowth activity suggesting that tumor innervation may be induced by tumor released exosomes in other solid tumors.

Our findings also indicate that other exosomal cargo including DNA, RNA, miRNA, and lipids should be examined for their axonogenic activity. Whatever the combination of factors involved, it is clear that interventions targeting tumor exosome release or blocking the ability of nerves to respond to exosomes may be of therapeutic value. While this concept requires rigorous testing, should it be proven valid, there would be great interest in its clinical translation.

Among the questions that require consideration is what advantage does innervation bestow on the tumor? Nerves generally bundle along with blood vessels, providing ready access to required nutrients. Thus, it is possible that tumors induce their own innervation to provide a rich blood supply and promote tumor growth. Our data support this hypothesis. Consistent with this, EphrinB1 possesses proangiogenic properties[62,63]. Alternatively, tumor innervation may regulate the local immune response which in many solid cancers, particularly head and neck cancers, critically contributes to disease progression[64]. Importantly, neuro-immune interactions are evolutionarily conserved and critical for homeostasis. Thus, tumors may promote their own innervation as a means to dampen immune responses, promote tumor tolerance, disease progression, and dissemination. Examining the relationship between tumor innervation, vascularization and immune infiltrates in the tumor will help distinguish between these possibilities.

## Methods

**Antibodies utilized for western blot analysis.** anti-CD9 (Abcam, ab92726, 1:1,000), anti-CD81 (clone B-11, sc-166029, 1:1,000, Santa Cruz), anti-EphrinB1 (ECD epitope, AF473, R&D Systems, 1:500), anti-EphrinB1 (ICD epitope, LifeSpan BioSciences, LS-C108001, 1:500), anti-NGF (Abcam, ab49205, 1:500), anti-β-III Tubulin (ab18207, Abcam, 1:1000), anti-GAPDH (ThermoFisher, AM4300, 1:5,000), anti-β actin (1:500, A2228, Sigma), anti-Tau (1:500, Abcam, ab-75714), anti-Phosphorylated Erk1/2 (1:500, #9106 s, Cell Signaling), anti-Erk1/2 (1:500, ABS44, Millipore), anti-TRPV1 (Novus, NB100-98897, 1:500). HRP- coupled secondary antibodies purchased from Thermo-Fisher. Citations for these validated antibodies are listed in Supplementary Table 9.

**Antibodies utilized for IHC.** anti-β-III Tubulin (2G10, ab78078, 1:250, Abcam), anti-Tyrosine Hydroxylase (Ab112, 1:750, Abcam), anti-TRPV1 (cat# ACC-030, 1:100, Alomone labs), anti-VIP (ab22736, 1:100, Abcam), anti-cytokeratin (Abcam, ab8068, 1:200).

**Antibodies utilized for immunofluorescence.** anti-Tau (MAB3420, 1:200, Millipore), anti-β-III tubulin (AB9354, 1:1500, Millipore), anti-TRPV1 (cat# ACC-030, 1:100, Alomone labs). Secondary antibodies: Goat anti Chicken IgY-568 (1:1500, Alexaflurr, Cat # A-11041), Goat anti mouse IgG2A-488 (1:1500, Alexafluor, Cat # A-21131), Goat anti Rabbit IgG-647 (1:1500, Alexafluor, Cat # A-21247).

**Antibody utilized for quantification of neurites.** anti-β-III tubulin (Millipore, AB9354, 1:1000).

**Antibody for NGF neutralization.** anti-NGF (ThermoFisher, PA1-18378, 6 µg/ml).

**Cell lines.** All human cell lines were authenticated by short tandem repeat profiling (BioSynthesis, Lewisville, TX or Genetica DNA Laboratories, Cincinnati, OH). In addition, all cell lines were confirmed mycoplasma free as per Uphoff and Drexler[65].

UM-SCC1, UM-SCC19, and UM-SCC47 cell lines maintained with DMEM with 10% fetal calf serum and 1% penicillin/streptomycin. The SCC1-*EphrinB1* cell line was generated by transfection of pcDNA3.1/Zeo-hEphrinB1 using Lipofectamine 2000 (ThermoFisher, #11668019) as per manufacturer's instructions. Following transfection, cells were treated with 250 µg/mL Zeocin

(ThermoFisher #R25005) until control cells were dead (7 days). Clones were isolated and expanded from 150 mm dishes seeded with approximately 50 cells using cloning cylinders (ThermoFisher #09-552-22) and screened by western blot for EphrinB1 expression (R&D Systems AF473).

Primary human tonsil epithelia were collected under an approved IRB protocol and maintained with KSFM (Gibco, cat # 10724-011). HTE *E6/E7* and HTE *E6Δ/E7* were generated by retroviral transduction and maintained in E-media as described below. Human squamous cell carcinoma cell (SCC) lines 1, 19 and 47 were a generous gift from Dr. Douglas Trask (University of Iowa, Iowa City, IA). These human cell lines were generated by the Head and Neck SPORE Translational Research Group at the University of Michigan (UM-SCCs; Ann Arbor, MI); the University of Michigan is where Dr. Trask originally obtained the cell lines. Completed genotyping of 73 UM-SCC cell lines has been published[66]. Continued efforts to genotype remaining and newly generated cell lines are posted on the UM Head and Neck SPORE Tissue Core web site (http://www2.med.umich.edu/cancer/hnspore/cores-tissue.cfm). The human 93-VU-147T-UP-6 squamous cell carcinoma cell line (referred to as 147 T in the text and figures) was a generous gift from Dr. Peter Snijders (Free University Hospital, De Boelelaan 1117, 1081 HV Amsterdam) and originated from the primary tumor of a patient that had HPV16 positive oral squamous cell carcinoma[67]. This cell line was maintained in the same medium as the other human SCC cell lines.

The CT26 cell line is a murine colorectal cancer cell line that was purchased from ATCC (CT26.WT, ATCC CRL-2638).

The B16 cell line is a murine melanoma cell line that was purchased from ATCC (B16-F0, ATCC CRL-6322).

The 4T1 cell line is a murine breast cancer cell line that was purchased from ATCC (ATCC CRL-2539).

mEERL cells (parental and all derivatives) were maintained with E-medium: DMEM (Corning, cat# 10-017-CV)/Hams F12 (Corning, cat#10-080-CV), 10% exosome depleted fetal calf serum, 1% penicillin/streptomycin, 0.5 µg/ml hydrocortisone, 8.4 ng/ml cholera toxin, 5 µg/ml transferrin, 5 µg/ml insulin, 1.36 ng/ml tri-iodo-thyonine, and 5 ng/ml EGF.

The mEERL *EphrinB1* CRISPR clones were generated using two distinct strategies. One strategy employed simultaneous double-targeting to remove a large portion of gDNA spanning exons 1-5 (as in ref. [68]) and one utilized single-targeting to produce frame-shift causing indels leading to early termination. Target selection and guide sequence cloning were carried out using the tools and protocol of Ran et al.[69]. PCR assays for the double targeting strategy employed primers external to (1-5Δ Ext.) or within (1-5Δ Int.) the predicted deletion site. The external assay should result in a 10,485 bp wt amplicon and a 229 bp Δ amplicon while the internal assay produces a 330 bp wt amplicon and no Δ amplicon. Single target screening utilized PCR to amplify a 330 bp region surrounding the target site followed by restriction digest with BslI, the recognition site of which should be destroyed when double strand breaks are incorrectly repaired. The primers sequence was as follow: -5Δ Ext. FWD sequence: 5′-ATCCTGAAGTGCATTCT GCC-3′; 1-5Δ Ext. REV sequence: 5′-TAGGGTACTGAGCGAGAGG-3′; 1-5Δ Int. FWD sequence: 5′-TGGCCTTCACTGTCATAGC-3′; 1-5Δ Int. REV sequence: 5′-TTCCAGGCCCATGTAGTTG-3′.

The mEERL *EphrinB1ΔECD* clone was generated from the double-targeting strategy. PCR assays show the predicted deletion product using primers external to the targeted region and lack of an amplicon using primers within the deletion (Supplementary Figure 4B). The sequence data shows that Exons 2-4 are deleted; these exons comprise the majority of the extracellular domain of *EphrinB1* (Supplementary Figure 4A). The 5′ end of the deletion in exon 1 occurs just after the signal peptide while the 3′ end of the deletion in exon 5 is within the transmembrane (TM) domain. Eight amino acids within the TM domain are deleted, however, two additional hydrophobic alanines are incorporated.

The mEERL *Rab27A$^{-/+}$Rab27B$^{-/-}$* clone resulted from a strategy for double knock out of *RAB27A* and *RAB27B* in which sgRNA's targeted to exon 4 of *RAB27A*, exon 4 of *RAB27B*, and a sequence of exon 3 shared by *RAB27A* and *B* were co-transfected. PCR revealed a heterozygous, truncated deletion product for *Rab27A* (Supplementary Figure 6A) and the expected homozygous deletion amplicon for *Rab27B* (Supplementary Figure 6B). *RAB27B* sequence data indicated distinct repair products at the deletion site; although one allele exhibits an immediate stop codon, it is unclear where the other might terminate. However, western blotting confirms lack of detectable protein (data not shown).

PC12 cells were purchased from ATCC and maintained with DMEM with 10% horse serum (Gibco, cat # 26050-088) and 5% fetal calf serum. When used for neurite outgrowth assays, PC12 cells were maintained with DMEM with 1% horse serum and 0.5% fetal calf serum.

**Electron microscopy.** Exosome samples were processed and analyzed by the Microscopy and Cell Analysis Core at Mayo Clinic: http://www.mayo.edu/research/core-resources/microscopy-cell-analysis-core/overview.

**Atomic force microscopy (AFM).** Purified exosomes were diluted 1:10 in deionized water, added to a clean glass dish, and allowed to air-dry for 2 h before drying under a gentle stream of nitrogen. Exosomes deposited on glass dish were characterized using an Atomic Force Microscope (Model: MFP-3D BIO$^{TM}$, Asylum Research, Santa Barbara, CA). Images were acquired in AC mode in air using a

silicon probe (AC240TS-R3, Asylum Research) with a typical resonance frequency of 70 kHz and spring constant of 2 Nm$^{-1}$. Height and amplitude images were recorded simultaneously at 512 × 512 pixels with a scan rate of 0.6 Hz. Image processing was performed using Igor Pro 6.34 (WaveMetrics, Portland, OR) and analyzed with Image J.

**Immunohistochemistry (IHC).** Tissues were fixed in 10% neutral buffered formalin and processed on a Leica 300 ASP tissue processor. All tissues were sectioned at 5 µm: $n = 15$ human HNSCCs stained for TH, VIP, and TRPV1 and $n = 30$ HNSCCs were stained for β-III tubulin. $n = 15$ mEERL tumors were stained or TH, VIP, and TRPV1 and $n = 30$ mEERL tumors were stained for β-III tubulin. $N = 12$ human HNSCC tumors were processed for double IHC to simultaneously label β-III tubulin and TRPV1. Double IHC was performed using MACH 2 Double stain 1 Polymer detection kit (Biocare, #MRCT523) as per manufacturer's instructions. The BenchMark® XT automated slide staining system (Ventana Medical Systems, Inc.) was used for the optimization and staining. The Ventana iView DAB detection kit was used as the chromogen and the slides were counterstained with hematoxylin. Omission of the primary antibody served as the negative control. Stained sections were analyzed on an Olympus BX51 upright microscope equipped with an Olympus DP71 color CCD camera; 20×/0.75 UPlanSApo and 40×/0.90 UPlan SApo objectives were used.

β-III tubulin/TRPV1 positive fibers were quantified by stereology on an Olympus BX50 upright microscope using Microbright Field Stereo Investigator (version 11.02) using the Cavalieri estimator. $N = 12$ formalin fixed, paraffin-embedded patient tumors were cut into 5 µm sections and double IHC stained for β-III tubulin and TRPV1 as described above. A grid size of 1500 µm×1500 µm was used with a counting frame of 100 µm×100 µm (optical fractionator probe). The number of single and double (TRPV1) positive β-III tubulin positive fibers were counted and the percent double labeled calculated for each sample.

**mEERL tumor immunofluorescent labeling and quantification.** Formalin fixed paraffin-embedded sections were deparaffinized as follows: 100% Histo-clear (National Diagnostics) for 5 min, 100% ethanol for 5 min, 90% ethanol for 5 min, 70% ethanol for 5 min and 5 min in PBS. Antigen retrieval was as follows: slides were incubated in 10 mM Sodium Citrate Buffer [2.94 g sodium citrate trisodium salt dihydrate ($C_6H_5Na_3O_7\cdot2H_2O$) to 1 L dH$_2$O; pH adjusted to 6.0 and 0.5 ml Tween 20 was added]. Slides were immersed in citrate buffer at 95 °C for 1 h, then allowed to cool on the bench top for 30 min. Slides were washed once in 1× PBS for 5 min. Sections were blocked in blocking buffer (1x PBS, 10% goat serum, 0.5% Triton-X 100) for 1 h at room temperature. Slides were then incubated in primary antibody overnight at 4 °C. Slides were then washed three times for 5 min each in 1x PBS and incubated in secondary antibody and Hoescht (1:1000, Invitrogen) at room temperature followed by three 5 min washes in 1x PBS. Coverslips were mounted with Faramount Mounting media, aqueous (Agilent). Stained slides were viewed on an Olympus FV1000 confocal microscope equipped with a laser scanning fluorescence and 12 bit camera; representative images shown. Images were taken at 512 × 512 pixels using the 60× oil PlanApo objective, integration type set at Line Kalman, integration count set at 2, LUT set at 0–3108. The zoom function was set at 2.6× and all images were processed the same.

**PC12 assay and β-III tubulin quantification by CX7.** The CellInSight CX7 High Content Analysis Platform performs automated cellular imaging for quantitative microscopy which was utilized to quantify neurite outgrowth. 7.5 × 10$^4$ PC12 cells were seeded onto 96-well black optical bottom, flat bottom plates (ThermoFisher) and 48 h after treatment were fixed with 4% paraformaldehyde and then blocked and permeabilized with a solution containing 3% donkey serum, 1% BSA, and 0.5% Triton-X 100. Staining for β-III tubulin (Millipore, AB9354) was followed by Alexa Fluor™ 488 goat anti-chicken IgG and Hoechst 33342. Washes were performed with PBS. Neurite outgrowth analysis was performed on the CellInsight CX7 HCS (ThermoFisher) using the Cellomics Scan Software's (Version 6.6.0, ThermoFisher) Neuronal Profiling Bioapplication (Version 4.2). Twenty-five imaging fields were collected per well with a 10× objective with 2 × 2 binning. Nuclei were identified by Hoechst -positive staining, while cell somas and neurites were identified by β-III tubulin -positive immunolabeling. Cells were classified as neurons if they had both a Hoechst -positive nucleus as well as a β-III tubulin positive soma. Only neurites longer than 20 µm were included in the analysis. All assays utilizing exosomes from cell lines were run with an $n = 3$–5 replicates per condition (based on exosome yield) and repeated at least two times (biological replicates) with similar results.

**NGF neutralization of mEERL conditioned media.** Fourty-eight hour conditioned media from mEERL parental and mEERL EphrinB1 cells was incubated with neutralizing anti-NGF antibody (6ug/mL) overnight at 4 °C. Conditioned media from $n = 5$ wells/condition were tested. Following incubation with the neutralizing antibody, the conditioned media (or control media) were used to stimulate PC12 cells as described.

**Exosome purification by differential ultracentrifugation.** 500,000 cells were seeded onto a 150 mm$^2$ plate and incubated in medium containing 10% fetal calf serum that was depleted of exosomes. Fetal calf serum exosome depletion consisted

of an over-night ultracentrifugation at 100,000×$g$. Conditioned medium was collected after 48 h and exosomes were purified by differential ultracentifugation as described by Kowal et al.[29] with some modifications. Briefly, conditioned medium was centrifuged at 300 × $g$ for 10 min at 4 °C to pellet cells. Supernatant was centrifuged at 2,000 × $g$ for 20 min at 4 °C, transferred to new tubes, and centrifuged for 30 min at 10,000 × $g$, and finally in a SureSpin 630/17 rotor for 120 min at 100,000 × $g$. All pellets were washed in PBS and re-centrifuged at the same speed and re-suspended in 200 µL of sterile PBS/150 mm dishes.

**Differential ultracentrifugation/optiprep density gradient.** Following differential ultracentrifugation as described above, a discontinuous iodixanol gradient was utilized similar to Van Deun et al.[36] with some modifications. Solutions of 5, 10, 20, and 40% iodixanol were made by mixing appropriate amounts of a homogenization buffer [0.25 M sucrose, 1 mM EDTA, 10 mM Tris-HCL, (pH 7.4)] and an iodixanol solution. This solution was prepared by combining a stock solution of OptiPrep™ (60% (w/v) aqueous iodixanol solution, Sigma) and a solution buffer [0.25 M sucrose, 6 mM EDTA, 60 mM Tris-HCl, (pH 7.4)]. The gradient was formed by layering 4 mL of 40%, 4 mL of 20%, 4 mL of 10% and 3 mL of 5% solutions on top of each other in a 15.5 mL open top polyallomer tube (Beckman Coulter). Four hundred microlitre of crude exosomes (isolated by differential ultracentrifugation) were overlaid onto the top of the gradient which was then centrifuged for 18 h at 100,000×$g$ and 4 °C (SureSpin 630/17 rotor, ThermoScientific™ Sorvall™). Gradient fractions of 1 mL were collected from the top of the self-forming gradient, diluted to 14 mL in PBS and centrifuged for 3 h at 100,000 × $g$ and 4 °C. The resulting pellets were re-suspended in 100 µL PBS and stored at −80 °C.

**Exosome purification from human plasma.** Twenty ml of whole blood were pipetted directly onto Ficoll-loaded Leucosep tubes and centrifuged at room temperature for 30 min at 800 × g with the brake off. Exosomes were isolated from the recovered plasma by differential ultracentrifugation as described.

**Exosome purification from human tumor.** Fresh tumor tissue was cut into small pieces and placed in culture with KSFM (keratinocyte serum free medium) containing Fungizone (Thermo Fisher) and maintained in culture for 48 h. Conditioned media was collected and exosomes harvested by differential ultracentrifugation as described.

**BCA protein assay of exosomes.** The standard BCA protein assay was utilized with modifications to accommodate the low protein yield from exosome preparations. Briefly, 5 µl of 10% TX-100 (Thermo Scientific) were added to an aliquot of 50 µl of purified exosomes and incubated 10 min at room temperature. A working ratio of 1:11 was used and incubated in a 96 well plate for 1 h at 37 °C. Absorbance at 562 nm was then measured (SpectraMax Plus 384) and protein concentration estimated from a quartic model fit to the BSA standard curve.

**Western blot analysis.** Sample protein concentrations were determined by BCA protein assay as described. Equal total protein was separated by SDS-PAGE, transferred to PVDF membranes (Immobilon-P, Millipore), blocked with either 5% Bovine Albumin Fraction V (Millipore) or 5% milk (Carnation instant non-fat dry milk), washed in TTBS (0.05% Tween-20, 1.37 M NaCl, 27 mM KCl, 25 mM Tris Base), and incubated in primary antibody. Washed membranes were incubated with HRP-conjugated secondary antibody, incubated with chemiluminescent substrate (ThermoScientific, SuperSignal West Pico) and imaged using a UVP GelDocIt 310Imaging System equipped with a high resolution 2.0 GelCam 310 CCD camera. Images were acquired using the VisionWorksLS Image Acquisition and Analysis Software with setting as follows: Aperture: F1.2; Focus: 56%; Lighting and filters: Clear and No illumination; Tray height: 90: Integration: Dynamic; Exposure time: 2 min; number of images: 20; Capture binning: 4 × 4.

For western blot analysis of neuritogenesis signaling in PC12 cells, 35 mm tissue culture dishes were coated with collagen (Sigma #5533) overnight at room temperature. The following day, excess collagen was aspirated and dishes washed with PBS prior to cell seeding. 1 × 10$^6$ PC12 cells were seeded on collagen-coated 35 mm dishes in complete media which consists of RPMI -1640 (Corning #10-040-CV) supplemented with 10% heat-inactivated horse serum (Gibco #26050088) and 5% fetal bovine serum (Tissue Culture Biologicals #101) overnight. All sera used was exosome depleted as described. The following day, the media was replaced with warm differentiation media (complete media diluted 1:10 in RPMI-1640) for 6 h at 37 °C. Cells were then washed one time with warm RPMI media and stimulated as follows (all stimulations were in RPMI media): 1) NGF (50 ng/ml), 2) PBS, 3) exosomes for 5 minutes. For each experimental condition, exosome inputs were normalized to 12.5 µg total protein as determined by BCA assay (ThermoFisher #23225), brought to a total volume of 100 µL in PBS, and added to 900 µL RPMI-1640. The NGF (Abcam #ab179616) positive control was diluted to 50 ng in 100 µL PBS while the negative control consisted of 100 µL of PBS alone; both were brought to a final volume of 1 mL with RPMI-1640. After stimulation, PC12 cells were harvested and lysed in lysis buffer (50 mM Tris-HCl pH 7.4, 100 mM NaCl, 100 mM NaF, 10 mM TSPP, 2 mM activated Na$_3$VO$_4$, 10% glycerol) with 1% Triton X-100 and 1X HALT protease inhibitor cocktail (ThermoFisher #78429). Following

BCA protein assay, 25 μg of whole cell lysate was separated by SDS-PAGE and analyzed by western blot.

Uncropped scans of key western blots can be found in Supplementary Figures 10–12.

**In vivo studies**. All animal studies were performed at Sanford Research which has an Animal Welfare Assurance on file with the Office of Laboratory Animal Welfare. The Assurance number is A-4568-01. Sanford Health is also a licensed research facility under the authority of the United States Department of Agriculture (USDA). The USDA certificate number is 46-R-009. The Sanford Health animal research program is also accredited by AAALAC, Intl. The animal facility is a specific pathogen free facility. All mice are maintained in IVC Tecniplast Green line Seal Safe Plus cages and cages are opened only under aseptic conditions in an animal transfer station. All cages are changed every other week using aseptic technique. Cages have individual HEPA filtered air. Animal rooms are maintained at 75°F, 30–70% humidity, have a minimum of 15 air changes per hour, and have a 14:10 light/dark cycle. Cages are maintained with corncob bedding and nesting material, both of which are autoclaved prior to use. Cages are maintained with irradiated, sterile food (Envigo) and acidified water (pH 2.8–3.0) available ad libitum. There is a maximum of 5 mice/cage. All animals are observed daily for abnormal behavior, signs of illness or distress, the availability of food and water and proper husbandry. All animal experiments were performed under approved Sanford Research IACUC protocols, within institutional guidelines and comply with all relevant ethical regulations. All animals injected with murine tumor cell lines utilized 4–8 week old male C57Bl/6 mice (The Jackson Laboratory). Animals injected with human tumor cell lines utilized 4–8-week-old male NOD-SCID mice (The Jackson Laboratory, 001303, $Prkdc^{scid}$).

Investigators were blinded to the groups when assessing animals (e.g. measuring tumors). Animals are housed in groups of 5 and are numbered by ear punch and cage number. No other identifiers (e.g. as to what group they are in) are on cages to maintain blindness of investigators. When measuring tumors, investigators do not have access to the identification key. When harvesting tumors for western blot quantification, investigators are no longer blinded to the groups to allow for appropriate gel loading of samples.

Tumors were initiated as follows: using a 23-gauge needle, cells ($1 \times 10^5$ cells) were implanted subcutaneously in the right hind limb of C57Bl/6 or NOD-SCID male mice. Tumor growth was monitored weekly by caliper measurements. $N = 4$ or 5 tumors/group were used for quantification of β-III tubulin by western blot. In these cases, mice were killed prior to tumor reaching killing criteria as noted in the text. $n = 7–10$ mice/group were used for tumor growth studies. Mice were euthanized when tumor volume was >1.5 cm in any dimension. Mouse numbers for each animal experiment are noted in appropriate figure legends.

**Whole tumor lysates**. Tumors were harvested post-implantation as indicated and homogenized in lysis buffer (50 mM Tris HCl pH 7.4, 100 mM NaCl, 100 mM NaF, 10 mM NaPPi, 2 mM Na$_3$PO$_4$, 10% glycerol, 1% Tx-100, HALT protease inhibitor cocktail) on ice using a tissue homogenizer (Omni TH International). The homogenate was then sonicated and centrifuged at 2000×g for 5 min. The resulting supernatant was collected and further centrifuged at 13,000 × g for 10 min prior to BCA protein concentration estimation. Western blots were conducted using 15 μg inputs.

Beta-III tubulin, TRPV1 and Tau western blots of whole tumor lysates were from $n = 4$ or 5 tumors/condition. Signals were normalized to GAPDH and quantified by densitometry using VisionWorks LS software.

**GW4869 treatment**. C57BL/6 male mice will be injected subcutaneously with $1 \times 10^5$ mEERL parental tumor cells on day 0. Beginning the following day (day 1), exosome release was systemically blocked by intraperitoneal injection of the neutral sphingomyelinase inhibitor GW4869 (dose of 1.25 mg/kg/day, Cayman Chemical) vs vehicle. Mice were treated 6 days a week, and administered a double dose (2.5 mg/kg) on the 6th day. GW4869 treatment continued for the duration of the experiment (27 days). A stock solution of 5 mg/ml of GW4869 was prepared in DMSO. The stock was diluted in PBS to appropriate concentration before injected into mice. Control mice were injected with diluted DMSO (vehicle). Blockade of exosome release by GW4869 treatment was assessed by collecting blood from all mice and purifying exosomes from the plasma. Exosomes were purified by differential ultracentrifugation as described. Purified exosomes were quantified by NanoSight.

**Human samples**. All human samples were collected under an approved Sanford Research Institutional Review Board protocol with signed Informed Consent and comply with all relevant ethical regulations. Samples included adult (age ≥ 18 years) patients of both sexes and all races with a diagnosis of primary or locally advanced, squamous cell carcinoma of the head and neck (anatomic sites: oral cavity, oropharynx, hypopharynx, and larynx). Children were excluded from this study as they do not suffer from HNSCC.

**Statistical analysis**. Data were analyzed and graphed using Graphpad Prism V6. Descriptive statistics are presented as mean ± standard deviation (see Figure legends). Two-tailed Student's *t*-test, one-way or two-way ANOVA were utilized for statistical analysis as indicated in the figure legends. Post hoc Fisher's Least Significant Difference (LSD) or Tukey test were used as noted in figure legends. All comparisons and LSD *p* values can be found in Supplementary Tables 1–8. PC12 assays utilizing exosomes from cell lines were run with three-five technical replicates (specific number of replicates is noted for each experiment in the relevant figure legends) for each condition and experiments were repeated at least 2 times (biological replicates). PC12 assays utilizing exosomes from human samples (plasma or tumor) were treated differently as these samples were limited. Thus, exosomes for each human sample were tested in triplicate; experiments repeated two times.

For animal studies, a statistical power analysis for a two-sample *t*-test using R software (version 3.4.1) was performed for sample size estimation as follows. For animal studies with $n = 4$ mice/group, we have more than 80% power to detect an expected effect size of 2.38 with an alpha = 0.05. For animal studies with $n = 7$ mice/group, we have more than 80% power to detect an expected effect size of 1.63 with an alpha = 0.05. For animal studies with $n = 10$ mice/group, we have more than 80% power to detect an expected effect size of 1.32 and an alpha = 0.05. Tumor growth curves were analyzed by two-tailed Student's *t*-test. Error bars are standard deviation or standard error of the mean (as indicated in figure legends).

For all datasets, qq plots were utilized to analyze normality. Any data points that were at least 1.5 interquartile ranges below the first quartile (Q1), or at least 1.5 interquartile ranges above the third quartile (Q3) of the data were considered outliers and were excluded from the analysis.

Tumor groups were assigned arbitrarily. Once tumors were initiated, mice were randomized to two groups. Randomization was performed by using the = Rand() function in Excel to generate a random number associated with each mouse; the lowest numbered mice were designated for one group (e.g. tumor growth analysis) and the higher numbered animals were utilized for another group (e.g. generating tumor lysate for western blot analysis). Two animal studies (NOD SCID mice for growing human SCC tumors, Fig. 7d–g, and GW4869 mouse experiment, Fig. 5 f–h) were utilized only for generating tumor for innervation analysis (whole tumor lysate or FFPE for IHC). Thus in these experiments, tumor types were assigned arbitrarily to mice but there was no randomization since all mice were sacrificed for the same purpose.

## Data availability
The data that support the findings of this study are available from the corresponding author upon reasonable request.

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

## Acknowledgements

This project was supported by an Institutional Development Award (IDeA) from the National Institute of General Medical Sciences of the National Institutes of Health under grant numbers 2P20GM103548 (Cancer), 5P20GM103620 (Pediatrics) and P20GM121341 (Population Health). Specifically, the Molecular Pathology Core, Imaging Core, and COMMAND (Collection Methods, Management and Analysis of Data) Cores provided their services and expertize towards this project. RD was supported by NIH (R01 CA193522), the NIH MD Anderson Cancer Center Support Grant (P30 CA016672) and an internally funded grant mechanism (IRG26881).

## Author contributions

M.M.: experimental conception, design, acquisition, analysis and interpretation of data, intellectual contributions, and critical review of manuscript. P.L.C.: experimental conception and design, acquisition, analysis and interpretation of data, intellectual contributions, and critical review of manuscript. D.W.V.: formulation of theory and prediction, experimental conception and design, acquisition, analysis and interpretation of data, intellectual contributions, and critical review of manuscript. C.T.L.: acquisition, analysis and interpretation of data, and critical review of manuscript. J.T.C.: experimental conception and design, acquisition and analysis of data. E.G.V.: acquisition, analysis and interpretation of data, and critical review of manuscript. A.J.G.: acquisition and analysis of data. D.M.: analysis of data. A.P.R.: acquisition, analysis and interpretation of data. Z.H.: analysis and interpretation of data. W.C.S.: experimental conception and design, intellectual contributions, and critical review of manuscript. J.Z.: statistical analysis of data. J.M.W.: intellectual contributions, and critical review of the manuscript. J.H.L.: intellectual contributions and critical review of manuscript. R.D.: analysis and data interpretation, intellectual contributions, and critical review of manuscript. P.D.V.: formulation of theory and prediction, intellectual contributions, experimental conception and design, drafting and revising of manuscript.

## Additional information

**Competing interests:** The authors declare no competing interests.

