## [Peer Review File · Nature Communications]

Reviewers' Comments:

Reviewer #1:

Remarks to the Author:

General comment:

This manuscript examines how cancer cell-derived exosomes induce tumor growth. The authors show that exosomes released by tumors can promote innervation, and depletion of exosomes inhibits innervation and reduces cancer growth. Although it is already known that cancer exosomes can promote tumor development, metastasis and drug resistance, the connection with tumor innervation is novel.

The main problem of the paper, however, is that the data are correlative; the link between innervation and tumor growth is not established. The authors show that exosomes promote innervation and exosome deficiency slows down disease progression, but direct evidence indicating that cancer exosome-induced innervation regulates tumor growth would be needed to justify the title and the main conclusion of the paper.

Other comments:

1. Additionally, the authors did not characterize the sensory nerve innervation in sufficient details. The authors showed that cancer derived exosomes promote neurite growth in vitro in Figs 1-2 by IHC staining and P12 neurite counts. However, in Fig4 and Fig 5, Western blot for tubulin -III is the only method used to characterize the innervation in vivo, which is not sufficient and quantitative. Immunofluorescence and quantification of tubulin -III and TRPV1 would be needed. It would also be more accurate to use both markers and quantify the abundance of TRPV1 positive nerve within the total Tubb3+ nerves.
2. The Western blot result in Fig4A shows that EphrinB1 and EphrinB1 ECD tumors express higher tubulin -III, as compared to parental. This is not consistent with in vitro data presented in Fig 2B, in which exosomes from EphrinB1, but not EphrinB1 ECD, can induce neurite outgrowth of P12 cells. In Fig4B, it seems that there is a trend that Null2 also increased tubulin -III expression. Data from more mice would be needed to have a more definitive conclusion.
3. The term "neo-neurogenesis" may not be appropriate here as it is usually reserved for the generation of neurons from a precursor cells. "Axonogenesis" would be more appropriate in the title and the text.
4. Formatting and methodological issues: Error bars appear to be missing in some panels (e.g. Fig1F and Fig1G, Fig5B and Fig5C). Scale bar is missing in Fig2C. In Fig4B, there is a star between ECD and Null2: is this significant? In statistical analyses, correction should be used when comparing multiple groups. In Fig 2E, the first lane for EphrinB1-In, a band is not clearly visible. There is no reference to Fig2C in the manuscript. Fig5E should be analyzed statistically.

Reviewer #2:

Remarks to the Author:

Innervated tumors tend to be more aggressive and metastatic, compared to non-innervated tumors, suggesting that the presence of neuronal-like cells within a tumor may promote disease progression. However, the mechanisms that underlie tumor innervation, as well as precisely how this process contributes to cancer cell aggressiveness, are not well understood. This manuscript describes a potentially novel way that tumor innervation might occur. Specifically, the authors show that primary head and neck tumor cells, and papillomavirus-induced oropharyngeal squamous cell carcinoma cells (referred to as mEERL cells), generate exosomes, a specific class of extracellular vesicles involved in intercellular communication, that are capable of inducing neurite outgrowth in PC12 cells. The authors then set out to determine how these effects were mediated.

They found that EphrinB1, a ligand that activates the Eph family of receptor tyrosine kinases, was expressed in cancer cell-derived exosomes, and stimulates the differentiation of PC 12 cells into neuronal-like cells. Consistent with this idea, ectopic expression of EphrinB1 in mEERL cells increased the ability of exosomes derived from these cells to promote neurite outgrowth, while depleting the cells of EphrinB1 blocked this effect. Lastly, the authors extended their study to include the use of a mouse cancer model to further substantiate their findings. However, the results from these in vivo experiments do not entirely recapitulate the cell-based findings.

Exosomes represent a unique form of intercellular communication which have been attracting a good deal of attention, especially in the cancer biology field. Thus, the findings from this manuscript showing that exosomes from certain types of cancer cells may promote tumor innervation are potentially interesting and important. The authors do a nice job generating homogenous preparations of exosomes, and provide some initial evidence suggesting that exosomes containing EphrinB1 might be involved in promoting innervation (i.e. these exosomes induced neurite outgrowth in cultures of PC12 cells). However, there are some issues with the manuscript that somewhat diminish my overall enthusiasm for the study. My feeling is that the study would require significant revision before it can be considered again for publication. Perhaps foremost among these issues are the results of the in vivo experiments shown in Figure 4. Here, the tumors formed by mEERL cells ectopically expressing the EphrinB1 delta-ECD mutant are clearly more innervated, compared to the parental cell line. However, the exosomes collected from cultures of these same cells are unable to promote neurite outgrowth in PC 12 cells, raising the question of just how important/necessary are exosomes and EphrinB1 for mediating tumor innervation? The authors attempt to address this inconsistency by suggesting that other factors in the tumor microenvironment are likely responsible for the innervation observed with this cell line; however, this does not alleviate the concerns. A better explanation of this result is required, in order to prove the contention of how tumor cell innervation is being achieved. The EphrinB1 null 2 mEERL cancer cell line appears to be giving rise to tumors containing more neuronal-like (i.e. beta-III tubulin positive) cells, compared to control cells (Fig 4B), which also requires some explanation.

Moreover, while collecting exosomes from the plasma and tumors of three head-and-neck cancer patients and showing that they can promote neurite outgrowths in PC12 cells, is a good start toward demonstrating that exosomes containing EphrinB1 are major contributors to tumor innervation, ideally, one would like to see exosomes isolated from a larger number of tumors (both innervated and non-innervated samples) to be examined for their ability to promote neurite outgrowth, and to relate these effects to differences in EphrinB1 expression levels in cells and exosomes. The manuscript would also benefit from some additional experiments: A few of the more obvious examples that come to mind include; 1) showing the expression levels of EphrinB1 in each of the cell lines/tissue samples used in the study, 2) showing that exosomes containing EphrinB1 can activate signaling events in PC 12 cells (and other neuronal cell types), that promote differentiation, while exosomes depleted of EphrinB1 cannot, and 3) determine the relative contributions of exosomes versus the other soluble factors secreted by cancer cells (i.e. NGF) in promoting tumor innervation to convincingly show that exosomes do indeed play a significant role in this process.

Lastly, the authors provide some experimental evidence suggesting that exosomes mediate tumor innervation which thereby increases tumor growth (Fig. 5). These findings are intriguing, but the authors might consider trying to further establish their point by showing that injecting poorly innervated tumors with exosomes derived from cancer cells, that are known to strongly stimulate neurite outgrowth in PC 12 cells, results in tumor innervation. I realize that this could be challenging, and thus would not insist on this, but if successful, such experiments would make a very powerful case for the authors' conclusions.

Reviewer #3:

Remarks to the Author:

In this article the authors propose that exosomes from HVP-positive head and neck cancers promote tumor innervation, which in turn increases tumor growth. Furthermore, they implicate the transmembrane ephrin, ephrinB1, in this activity of exosomes. This is an interesting and important area of investigation.

The conclusions that exosomes derived from head and neck cancer cells promote neurite outgrowth in cultures of PC12 pheochromocytoma cells and that ephrinB1 is present in the exosomes and plays a role in this activity seem well supported by the data. However, a number of the findings reported are somewhat preliminary. More evidence is needed to verify some of the conclusions and yield insight into the mechanisms underlying the observed phenomena, so that a coherent model for the role of exosomes and ephrinB1 in tumor innervation can be outlined.

Some of the issues that should be addressed include the following.

Beta-III tubulin expression does not conclusively demonstrate tumor innervation. Beta-III tubulin has been reported to be present in many types of cancer cells, including head and neck cancers, and to correlate with malignancy and resistance to chemotherapeutic drugs targeting microtubules. There are also reports of TRPV1 expression in cancer cells, and controls should be included to verify the specificity of the TRPV1 antibodies. In addition, the authors should perform double labeling for beta-III tubulin and TRPV1 to show that they are colocalized in the "nerve twigs" (the staining patterns for the two proteins do not look similar in the images provided in Figs. 1A and 2A). In addition, the authors should also show colocalization of beta-III tubulin/TRPV1 with additional neuronal-specific markers. These data would eliminate the possibility that head and neck cancer exosomes/ephrinB1 increase tumors malignancy by other mechanisms, which may lead to increased beta-III tubulin expression by the tumor cells independently of innervation.

It would also be valuable to determine whether the "neural twigs" express Eph receptors that can respond to ephrinB1.

Ephrins have been linked to tumor angiogenesis. Experiments should be performed to exclude that the growth-promoting effects of exosomes bearing ephrinB1 in head and neck tumors are linked to increased angiogenesis rather than (or in addition to) innervation.

It is not clear how ephrinB1 lacking the extracellular region (ephrinB1 deltaECD) can still promote tumor innervation in vivo. Ephrin-B1 is a ligand that activates EphB receptor tyrosine kinases, an activity that requires the extracellular region. The authors speculate that a binding partner capable of interacting with both full-length ephrinB1 and ephrinB1 delta ECD could be responsible for neurite outgrowth/innervation induced by the exosomes, rather than ephrinB1 itself. Experiments should address this possibility. In addition, the truncated ephrin-B1 deltaECD did not potentiate neurite outgrowth in cultured PC12 cells (used to assay neurite-promoting effects of exosomes). The authors speculate that the tumor microenvironment may be involved in the in vivo tumor innervation by truncated ephrin-B1, but no evidence is presented to support this speculation and/or a potential mechanism.

In Fig. 2E, the ephrinB1-In antibody labels bands of similar size in cells expressing wild-type ephrinB1 and ephrinB1 deltaECD. Shouldn't ephrinB1 deltaECD be smaller in size than full-length ephrinB1?

Did the mEERL tumors with high ephrinB1 or ephrinB1 delta ECD expression (Fig. 4) both exhibit faster growth as compared to control tumors, correlating with the similarly increased levels of beta-III tubulin induced by the two forms of ephrinB1? Data shown in the manuscript demonstrate that production of exosomes is important for tumor growth, but not necessarily that ephrin-B1 is required for the growth promoting activity of the exosomes.

The data in Fig. 3 are disconnected from the rest of the manuscript. How is the role of HPV16 in promoting neurite outgrowth connected with that of ephrinB1? What are the levels of ephrinB1 in the four head and neck cancer cell lines examined? Can ephrinB1 overexpression increase the neurite outgrowth promoting activity of exosomes from HPV-negative cancer cells? Alternatively, is HPV16 required for the effects of ephrinB1?

In Fig. 1E, exosomes from Pt2 and Pt3 tumor tissue do not express detectable ephrin-B1, yet they promote neurite outgrowth in PC12 cells. This does not seem consistent with the proposed important role of ephrinB1 in the neurite-inducing effects of patient exosomes.

Additional controls are needed to determine whether the effects of Rab27A and B gene inactivation on tumor growth are due to deficient generation of exosomes or other possible activities of the Rab27 proteins in cancer cells.

It would be interesting to know whether the effects observed are specific for ephrin-B1 or whether the related ephrin-B2, which is also highly expressed in many cancer cells, is also involved.

Response to reviewers

Reviewer #1

1) Data is correlative; the link between innervation and tumor growth is not established. The authors show that exosomes promote innervation and exosome deficiency slows down disease progression but DIRECT evidence indicating that cancer exosome-induced innervation regulates tumor growth would be needed to justify the title and main conclusions of the paper.

The reviewer makes a good point. The experiments required to make the claim that tumor innervation contributes to tumor growth are beyond the scope of this manuscript even if they are part of our agenda for the future. Thus, we have revised the title to eliminate the link between innervation and tumor growth. The title now reads “Cancer exosomes induce tumor innervation.” The main conclusions of the manuscript have likewise been changed. We want to stress that, like the first version of the manuscript, the revised version of this manuscript is focused on tumor exosome-mediated innervation. We have made changes in the text and in the order of data presentation to better make this point. Moreover, tumor growth curves have been moved to supplemental data to further indicate that they are not the focus of the study and a direct link between tumor innervation and tumor growth is not being made by this study.

2) The authors did not characterize the sensory nerve innervation in sufficient detail.

We now include double IHC of human tumors for β -III tubulin and TRPV1 (Figure 1C). FFPE blocks of patient tumors were double stained by IHC and representative images demonstrating co-localization of β -III tubulin and TRPV1 are included (Figure 1C). β -III tubulin/TRPV1 positive fibers were quantified by stereology using Microbright Field Stereo Investigator (version 11.02) and the Cavalieri estimator. N=12 formalin fixed, paraffin-embedded patient tumors were cut into 5 μ m sections and double IHC stained for β -III tubulin and TRPV1 as described above. A grid size of 1500 μ m x 1500 μ m was used with a counting frame of 100 μ m x 100 μ m (optical fractionator probe). The number of single (β -III tubulin) and double (β -III tubulin and TRPV1) positive fibers were counted and the data are included in Figure 1D which clearly shows that the majority of β -III tubulin positive fibers are also TRPV1 positive. When averaged across all the samples, 83.7% (\pm 6.5) of β -III tubulin positive nerve twigs are also TRPV1 positive indicating that the overwhelming majority of single innervating fibers are sensory in nature. **These changes can be found in the text on page 3, lines 94-103.** In addition, we now also include triple immunofluorescence of mEERL tumors for β -III tubulin, tau (both neuronal markers) and TRPV1 (sensory marker) (Figure 4C). **These changes can be found on page 4, lines 137-151.**

3) The authors show that cancer derived exosomes promote neurite outgrowth in vitro in Figs 1-2 (now figures 3 and 5) by IHC staining and PC12 neurite counts. However, in Fig 4 and Fig 5, western blot for tubulin-III is the only method used to characterize in vivo, which is not sufficient and quantitative. Immunofluorescence and quantification of tubulin-III and TRPV1 would be needed. It would also be more accurate to use both markers and quantify the abundance of TRPV1 positive nerve within the total Tubb3+ nerves.

For in vivo tumor analysis of innervation, western blot analysis of whole tumor lysate was chosen as it provides a quantitative measure of innervation of the entire tumor. IHC provides a visual account of nerve fibers within tumor tissue. However, even when quantified, it represents neural elements present in that slice (or slices) which may not be representative of the extent of innervation in the tumor as a whole. If hotspots of innervation exist, the data could be skewed if analyzed solely by IHC/IF. Thus, we have included both western blot and IHC/IF analysis of tumor innervation. We now present western blot analysis of whole tumor lysate for β -III tubulin, TRPV1 and Tau (another neuronal marker). Please see Figures 5B, 7D, and Supplemental Figure 8A for western blots. **These changes can be found on page 5, lines 187-197; page 7, lines 304-308 and continued on page 8, lines 309-313.** In addition, we double IHC labeled patient tumors and quantified the

percent of β -III tubulin that are also TRPV1 positive. **This change can be found on page 3, lines 94-103.** Please see response to question #2 for details. Taken together, these data demonstrate qualitatively and quantitatively that mEERL tumors are innervated predominantly by sensory nerves *in vivo*.

4) The western blot result in Fig 4A (now Supplemental Figure 8B) shows that EphrinB1 and EphrinB1 ECD tumors express higher tubulin-III as compared to parental. This is not consistent with *in vitro* data presented in Fig 2B (now Fig 6C), in which exosomes from EphrinB1, but not EphrinB1 ECD, can induce neurite outgrowth of PC12 cells. In Fig 4B (now Supplemental Figure 8B), it seems that there is a trend with Null2 also increased tubulin-III expression. Data from more mice would be needed to have a more definitive conclusion.

We would like to point out that the PC12 assay shows that EphrinB1 Δ ECD exosomes *do* have significant neurite outgrowth activity above that of the negative control (PC12 cells alone) (Figure 6C). This is also true for Null 1 and Null 2 exosomes. However, we agree with the reviewer that the **extent** of *in vivo* innervation evident in the EphrinB1 Δ ECD tumors is higher than would have been predicted based on the PC12 assay. This was pointed out by all reviewers. Thus, we repeated the *in vivo* experiment as suggested, increasing the N to 10 mice/group. The new data are now presented in Supplemental Figure 8 and are similar to the original data. The data show that EphrinB1 Δ ECD tumors are innervated similarly to mEERL parental tumors (Supplemental Figure 8A-D). Since the data are reproducible, we believe this is, in fact, the case. It is important to keep in mind that, while useful for quantifying neurite outgrowth potential of exosomes, the PC12 assay is just that, an *in vitro* assay and, as such, it lacks many components that likely influence tumor innervation *in vivo*. However, both the *in vitro* PC12 assay and the *in vivo* experiment support the same conclusions:

- a) Increased EphrinB1 potentiates innervation.
- b) The lack of full length EphrinB1 expression (Δ ECD, null 1, null 2) does not eliminate innervation, further indicating that EphrinB1 is NOT required for innervation (but, instead, acts as an important potentiator of innervation).

These conclusions, together with the contribution of tumor released exosomes to tumor innervation, are the main conclusions of this work and are supported by the other data in the manuscript. **These changes can be found on page 8, lines 314-331.**

5) The term “neo-neurogenesis” may not be appropriate here as it is usually reserved for the generation of neurons from precursor cells. “Axonogenesis” would be more appropriate in the title and text.

We agree with the reviewer and have made the requested change.

6) Error bars in figures 1F (now Figure 3), 1G (now Figure 3), 5B (now Supplemental Figure 3B) and 5C (now Supplemental Figure 3C) are missing.

As mentioned in the original text, Figures 1C and 1F lack error bars because the exosomes purified from these samples (N11, Pt1-plasma, Pt1-tumor, TL) were low and allowed only for one replicate in the experiment. That is why they did not have error bars. Since the exosome yields from all other human samples (both plasma and tumor) were sufficient for N=3 biological replicates (and thus have error bars), and we were able to increase the N of patient samples analyzed, we decided to eliminate Pt1 and N11 data (samples have thus been re-numbered to avoid confusion). We have, however, kept TL1 data (despite the low exosome yield) as this is the only control tissue in that experiment (Figure 3). All experiments were replicated at least 2 times. **These changes can be found on page 3, lines 118-129 and continuing on page 4, lines 130-136.**

The error bars in Supplemental Figures 3B and 3C have been fixed.

7) Scale bar is missing in figure 2C.

Given the increased amount of data included in the manuscript to address reviewers' comments, we decided to eliminate Figure 2C as it was merely an image of PC12 cells after treatment and did not add any additional information to the data that are already in the manuscript.

8) In Fig 4B, there is a star between ECD and Null 2. Is this significant?

This experiment has been repeated and the old data replaced with the new data (Supplemental Figure 8). We show western blot quantification of whole tumor lysate for β -III tubulin, Tau and TRPV1 (Supplemental Figures 8B, 8C, 8D). There is no star between deltaECD and Null 2 in any of these quantifications (i.e. they are not significantly different). **These changes can be found on page 8, lines 314-331.**

9) In statistical analyses, correction should be used when comparing multiple groups.

The Benjamini-Hochberg procedure was used to adjust p values for all experiments where multiple comparisons are made. The adjusted p values are reported and noted as such in the appropriate figure legends.

10) In Fig 2E (now Figure 6D), the first lane for EphrinB1-In, a band is not clearly visible.

The western blot using the EphrinB1-In antibody has been repeated. It should be noted that full length EphrinB1 is naturally processed. It is first cleaved extracellularly by MMPs and then intracellularly by the gamma secretase complex. This intracellular cleavage releases a small fragment of approximately 15kD. This fragment is present in lysates from mEERL parental, EphrinB1 as well as EphrinB1 Δ ECD and is detected by the EphrinB1-In antibody. **This change can be found on page 6, lines 256-262.**

11) There is no reference to Fig 2E (now Figure 6D) in the manuscript.

Figure 6D is referenced in the results section titled "Exosomes induce neurite outgrowth without NGF." We have added an additional reference to Fig 6D to emphasize the inclusion of the intracellular domain in exosomes from the delta ECD cells. The text reads as follows:

"Consistent with the published literature, EphrinB1 is indeed packaged within exosomes as assessed by western blot analysis (Figure 6D) (35) (Exocarta.org). Moreover, while the extracellular domain of EphrinB1 is absent in mEERL EphrinB1 Δ ECD exosomes, the intracellular domain remains as cargo running at a similar size to (endogenously) proteolytically processed EphrinB1 (Tomita T et al. *Molecular Neurodegeneration* 2006, 1:2) (Figure 6D)." **This change can be found on page 6, lines 256-262.**

12) Error bars appear to be missing in Fig 1F and 1G (now Figure 3).

Please see the answer to question #6.

13) Figure 5E (now Supplemental Figure 3D) should be analyzed statistically.

The tumor growth curves have been analyzed statistically and p value has been added to the figure.

Reviewer #2

1) Perhaps the foremost among these issues are the results from the in vivo experiments in Figure 4 (now Supplemental Figure 8). Here, the tumors formed by mEERL cells ectopically expressing the EphrinB1 deltaECD mutant are clearly more innervated, compared to the parental cell line. However, the exosomes collected from cultures of these same cells are unable to promote neurite outgrowth of PC12 cells, raising the question of just how important/necessary are exosomes and EphrinB1 for mediating tumor innervation?

This is a similar concern raised by Reviewer 1. Please see our response to question 4 for Reviewer 1.

To summarize, we have added the following data to address the importance/necessity of exosomes and EphrinB1 in mediating tumor innervation. 1) NGF neutralization of conditioned media does not attenuate

neurite outgrowth of PC12 cells in vitro, 2) *treatment of tumor bearing mice with GW4869, an exosome release blocker, significantly decreases tumor innervation*, 3) *exosomes (regardless of their EphrinB1 status) induce activation of signaling consistent with neuritogenesis and 4) over-expressing EphrinB1 in a squamous cell carcinoma cell line significantly increases neurite outgrowth activity of its exosomes in vitro and tumor innervation in vivo*. Taken together these data clearly demonstrate the importance of exosomes in axonogenesis and EphrinB1 as potentiating this activity. Below are the details of the mentioned experiments.

To address the question of how important or necessary are exosomes to mediating tumor innervation, we have added the following data. To address the role of exosomes in neurite outgrowth, we now include data demonstrating that neutralization of NGF from conditioned media of mEERL cells has no effect on neurite outgrowth (Figure 6A). These data indicate that neurotrophic factors are not required for this activity; NGF is the only neurotrophic factor expressed by mEERL cells (Geo Accession GSE68935)(Vermeer, et al *Oncotarget* 7, 24194-24207 (2016). Moreover, we demonstrate that NGF is not packaged within exosomes (Figure 6B). Thus, factors present in the conditioned media (other than neurotrophic factors) harbor neurite outgrowth activity. Our data indicate that these factors are exosomes. **This change can be found on page 6, lines 223-237.**

We have also added another experiment indicating a significant role of exosomes in tumor innervation. We have utilized GW4869 in mouse studies. GW4869 is a cell permeable, neutral sphingomyelinase inhibitor that blocks exosome release. GW4869 blocks the ceramide-mediated secondary invagination of endosomes that generates the multi-vesicular body (MVB) from which exosomes are released. In this way, GW4869 inhibits release of mature exosomes from MVBs. In our experiment, mice were given mEERL parental tumors (1X10⁵ cells injected/mouse) and segregated into one of two groups (n=7 mice/group). Beginning the day after tumor inoculation, mice received daily intraperitoneal injections of GW4869 (dose = 1.25 mg/kg/day) vs vehicle. Mice were treated 6 days a week, and administered a double dose (2.5 mg/kg) on the 6th day. Blood was collected, exosomes purified from the plasma as described and quantified by nanoparticle tracking analysis. The data demonstrate a significant reduction in exosome release (Figure 5F) indicating that GW4869 worked. At end point, tumors were harvested, paraffin-embedded and IHC stained for β -III tubulin. Stained tumors were scanned on an Aperio ImageScope slide scanner, images exported and quantified using ImageJ software. Figures 5G and 5H show that tumors from GW4869 treated mice had significantly less β -III tubulin positive fibers as compared to those from vehicle treated mice. These data clearly support a role for exosomes in tumor innervation. **This change can be found on page 5, lines 203-219 and continuing on page 6, lines 220-222.**

We have also added new data supporting a role of exosomes in activating signaling pathways (MAP Kinase) known to be important for axonogenesis. We now show that mEERL exosomes induce MAP kinase signaling in PC12 cells (Figure 6E). We show that this signaling is activated regardless of the presence or absence of EphrinB1 in exosomes, consistent with our other data indicating that exosomal EphrinB1 is not required for neurite outgrowth activity. The data emphasize the point that *exosomes* activate signaling pathways that drive axonogenesis. While this signaling may be induced via receptor binding, our data suggest that PC12 expressed receptors are *not* binding exosomal EphrinB1 but rather another, as yet unidentified, exosomal ligand(s). Alternatively, exosomal membranes can fuse with the plasma membrane of PC12 cells and, in doing so, dump their cargo into the PC12 cell cytosol which may then initiate MAP kinase signaling. Finally, exosomes can be endocytosed by PC12 cells and, in this way, initiate MAP kinase signaling. Regardless of the mechanism, our data support a role for exosomes in mediating signaling cascades known to be important for axonogenesis. **This change can be found on page 7, lines 263-275.**

Our data indicate that, while not required for neurite outgrowth activity, EphrinB1 potentiates this activity. Thus, to address the question of how important or necessary EphrinB1 is to axonogenesis, we have added the following new data. To further address its role in tumor innervation, we over-expressed EphrinB1 in the HPV negative human SCC1 cell line. We now show that the SCC1-EphrinB1 exosomes potentiate neurite outgrowth

of PC12 cells as compared to neurite outgrowth of parental SCC1 exosomes (Figure 7C). Moreover, when injected into NOD-SCID mice, SCC1-EphrinB1 tumors are more innervated as assessed by western blot for β -III tubulin, TRPV1, and Tau (Figure 7D-G). Taken together, the data clearly support the concept that exosomal EphrinB1 potentiates neurite outgrowth activity. **This change can be found on page 7, lines 300-308 and continuing on page 8, lines 309-313.**

2) While collecting exosomes from the plasma and tumors of three head and neck cancer patients and showing that they can promote neurite outgrowth of PC12 cells is a good start toward demonstrating that exosomes containing EphrinB1 are major contributors in tumor innervation, one would like to see exosomes isolated from a larger number of tumors (both innervated and non-innervated samples) to be examined for their ability to promote neurite outgrowth, and to relate these differences to difference in EphrinB1 expression levels in cells and exosomes.

The point of showing that patient plasma and tumor exosomes induce neurite outgrowth was not to emphasize a role for EphrinB1 in neurite outgrowth but rather to emphasize a role for exosomes in neurite outgrowth activity in human disease (irrespective of EphrinB1 status). In fact, exosomes from the tumors of Pts3, 6 and 7 lack EphrinB1 in their exosomes (Supplemental Figure 1D) yet show significant neurite outgrowth activity (Figures 3B). We now include six additional head and neck cancer patient (Pt3-8) tumor and plasma exosomes in the PC12 assay (Figure 3). **This change can be found on page 3, lines 118-129 and continuing on page 4, lines 130-136.**

We realize that the reviewer asked to include patient samples from both innervated and non-innervated tumor samples. We must stress, however, that the tumors we obtain for these studies are not ones that we pick and choose based on any criteria other than that they are large enough to allow for a small piece to be given to the laboratory for this project. Moreover, these samples come to the laboratory directly from the operating room without any information (e.g. innervation status) other than the site of disease. Thus, we cannot choose innervated and non-innervated tumor samples to use in this study. Basically, we get what we get. In addition, most head and neck tumors are generally not large enough to allow for tumor tissue to come to the laboratory as most of the sample is needed for pathological evaluation and treatment decision-making, neither of which can be compromised for research purposes. Thus, increasing the N of matched blood and tumor samples for this study was not trivial.

The reviewer asks us to relate differences in EphrinB1 expression levels in cells (tumor cells) and exosomes with differences in neurite outgrowth activity (*in vitro* assay with PC12 cells). We do not posit that high tumor EphrinB1 levels necessitate high EphrinB1 cargo in their corresponding exosomes. While we see this in our mEERL cell lines in which EphrinB1 is over-expressed, that does not necessarily mean this is the case in patient tumor tissue. In fact, the machinery utilized to package cargo within exosomes is not fully defined nor are the changes to this machinery associated with disease (cancer) understood. Moreover, we stress that EphrinB1 is **not** the key to exosome-mediated tumor innervation but rather functions as a potentiator. Despite these caveats, we include here (for the reviewers) qRT-PCR analysis for EphrinB1 from tumors of Patients 1-8. We stress that these data were generated using RNA extracted from the tumor which does not solely contain cancer cells but also immune cells, stroma, etc. Thus, we cannot say that the RNA is pure cancer cell RNA. With this caveat in mind, we attempted regression analysis to see if there is a correlation between tumor EphrinB1 expression and exosome mediated *in vitro* neurite outgrowth activity. These data are also included below. As one can see, there is no obvious correlation. We believe that while the PC12 assay can be indicative of the axonogenesis potential of exosomes, this potential may be quite different *in vivo*. In the PC12 assay, the only recipient cells able to respond to exosomes are the PC12 cells. *In vivo*, tumor released exosomes diffuse away from the tumor and may be taken up by many different cell types in their local environment, as well as more distally. With all these caveats in mind, it is not surprising that no correlation between tumor EphrinB1

expression and PC12 neurite outgrowth activity was found. These data are consistent with our main observation that EphrinB1 is not necessary for exosome-mediated neurite outgrowth activity.

EphrinB1 expression of HNSCC patients (Pt1-8) relative to control tonsil. Error bars represent one standard deviation based on technical replicates from a single experiment.

Linear regression analysis of ΔCt for patient EphrinB1 expression and neurite outgrowth activity (as measured by total neurite count/well in the PC12 assay). $R^2 = 0.168$, $F = 1.211$, $p = 0.313$.

3) Show expression levels of EphrinB1 in each of the cell lines/tissue samples in the study.

EphrinB1 western blot analysis can now be found in figures: 2D, 6D, Suppl Figure 1, Suppl Figure 2D, Suppl Figure 7A and Suppl Figure 9.

4) Show that exosomes containing EphrinB1 can activate signaling events in PC12 cells (and other neuronal cell types) that promote differentiation while exosomes depleted of EphrinB1 cannot.

We stimulated PC12 cells with either PBS (negative control), NGF (positive control), exosomes containing basal levels of EphrinB1 (purified from mEERL parental conditioned media), exosomes with high EphrinB1 (purified from mEERL EphrinB1 conditioned media) or exosomes with no EphrinB1 (purified from mEERL Null1 conditioned media). Stimulation was for 5 minutes after which time PC12 cells were harvested and their whole cell lysate analyzed by western blot. We now show that exosomes, *regardless* of their EphrinB1 status, induce Erk1/2 phosphorylation (Figure 6E). **This change can be found on page 7, lines 263-275.** The MAP Kinase pathway is activated during neurite outgrowth (Jessen et al, Journal of Neurochemistry 79: 1149-1160, 2001) (Vaudry et al, Science 296: 1648-1649, 2002). These data support our finding that *exosomes* are important in driving neurite outgrowth. While binding of EphrinB1 to receptors expressed on PC12 cells may induce MAP Kinase signaling, this binding event is clearly not required to initiate signaling down this pathway. Additional experiments are needed to address the mechanism(s) initiating signal transduction but are beyond the scope of this manuscript.

5) Determine the relative contributions of exosomes vs the other soluble factors secreted by cancer cells (i.e. NGF) in promoting tumor innervation to convincingly show that exosomes do indeed play a significant role in this process.

We now present data that neutralizing NGF (Figure 6A) from conditioned media of mEERL parental and EphrinB1 cells does not affect neurite outgrowth activity. These data indicate that this neurotrophic factor does not contribute to neurite outgrowth activity of conditioned media. NGF is the only neurotrophic factor expressed by mEERL cells (GEO Accession GSE68935)(Vermeer, et al *Oncotarget* 7, 24194-24207 (2016). Data in Figure 6C shows that exosomes purified from conditioned media of mEERL cells contain neurite outgrowth activity. Moreover, we show that NGF is not packaged in exosomes (Figure 6B) and thus does not account for exosome-mediated neurite outgrowth activity. **This change can be found on page 6, lines 223-237.**

6) Lastly, the authors provide some experimental evidence suggesting that exosomes mediate tumor innervation which thereby increases tumor growth (now Supplemental Figure 3D). These findings are intriguing, but the authors might consider trying to further establish their point by showing that injecting poorly innervated tumors with exosomes derived from cancer cells, that are known to strongly stimulate neurite outgrowth in PC12 cells, results in tumor innervation. I realize that this could be challenging, and thus would not insist on this, but if successful, such experiments would make a very powerful case for the authors' conclusions.

We agree with the suggested *in vivo* experiment of taking poorly innervated tumors and injecting exosomes to see if this can result in enhanced innervation. This is a rather complex experiment and one that we have proposed in a grant application. We proposed to test exosomes purified from mEERL EphrinB1 and null 1 cells in mice bearing mEERL EphrinB1 null 1 tumors (no EphrinB1 and low innervation). Given their difference in EphrinB1 cargo, the experiment would also test the contribution of EphrinB1 in potentiating tumor innervation. There are many things to consider with such an experiment. For example, how often to inject exosomes? A second question is the route of administration? IP? Intra-tumoral? Peri-tumoral? When to begin exosome injections? The day after tumor inoculation? Or is it best to allow time for the post inoculation inflammation to subside before beginning exosome injections? These are a few of the variables that must be tested in order to derive meaningful data from such an experiment. Thus, we agree with the reviewer, that this would be the "killer" experiment and could provide the strongest evidence for the role of exosomes in mediating tumor innervation. However, given the complexity of the experiment, we feel it is beyond the scope of this paper.

Reviewer #2 agreed on the challenging nature of this experiment and did not insist on it. Given this, we have de-emphasized the contribution of innervation to tumor growth. The manuscript text has been changed to stress the role of *exosomes* in inducing tumor innervation and not tumor growth. As such, all tumor growth curves have been moved to supplemental figures and we make no conclusions on the role of innervation on tumor growth.

Reviewer #3.

1) Beta-III tubulin expression does not conclusively demonstrate tumor innervation. Beta-III tubulin has been reported to be present in many types of cancer cells, including head and neck cancers, and to correlate with malignancy and resistance to chemotherapeutic drugs targeting microtubules. There are also reports of TRPV1 expression in cancer cells and controls should be included to verify the specificity of the TRPV1 antibodies.

We agree with the reviewer that beta-III tubulin expression alone does not conclusively demonstrate tumor innervation and that it can be present in the cancer cells themselves. We now include additional IHC of mEERL tumors showing that the β -III-tubulin positive nerve twigs we have identified within tumor tissue are very different from cytokeratin stained tumor cells (Figure 4B). **This change can be found on page 4, lines 143-151.** In addition, we also include western blot analysis of mEERL cells indicating that they do not endogenously express β -III-tubulin, Tau (another neuronal marker) or TRPV1 (Supplemental Figure 8F). **This change can be found on page 8, lines 328-331.** In addition, we also include co-IHC/IF of β -III tubulin (neuronal marker) and TRPV1 (sensory marker) for human (Figure 1C) and quantification (by stereology) of β -III tubulin fibers that are also TRPV1 positive ($83.7\% \pm 6.5$). **This change can be found on page 3, lines 94-103.** Please see the more detailed answer to the same question for Reviewer #1 (question 2). co-immunofluorescence of β -III tubulin, tau (neuronal markers) and TRPV1 (sensory marker for mEERL (Figure 4C) tumors was also performed. **This change can be found on page 4, lines 143-151.** Moreover, we include western blot analysis of whole tumor lysates for beta-III tubulin, TRPV1 and Tau (another neuronal marker) (Figures 5B-E, 7D-G, Supplemental Figure 8 A-D).

2) The authors should perform double labeling for beta-III tubulin and TRPV1 to show that they are co-localized in the nerve twigs.

We now include representative images as well as quantification of double IHC stained patient samples for beta-III tubulin and TRPV1 (Figure 1C). **This change can be found on page 3, lines 94-103.** We also include triple IF of mEERL tumors for β -III tubulin, tau (neuronal markers) and TRPV1 (sensory marker) (Figure 4C). **This change can be found on page 4, lines 146-151.**

3) The authors should show co-localization of beta-III tubulin/TRPV1 and additional neuronal specific markers.

We now include IHC of human tumors co-staining β -III tubulin and TRPV1 (Figure 1C). **This change can be found on page 3, lines 94-103.** We also include co-immunofluorescence of mEERL tumors for β -III tubulin, tau (neuronal markers) and TRPV1 (sensory marker) (Figure 4C). **This change can be found on page 4, lines 146-151.** We also include western blot analysis of whole tumor lysates for beta-III tubulin, TRPV1 and Tau (Figures 5B-E, 7D-G, Supplemental Figure 8A-D).

4) It would also be valuable to determine whether the neural twigs express Eph receptors that can respond to EphrinB1.

While we agree with the reviewer, the lack of expression of Eph receptors on neural twigs would not indicate that they are incapable of responding to EphrinB1. EphrinB1 is a promiscuous ligand that can interact with

several non-Eph receptors including FGFR, IL7-R, EGFR, ErbB2, ErbB3, ErbB4 (Vermeer et al, Cancer Research 73: 5787-5797, 2013)(Lee et al, Molecular Biology of the Cell 20: 124-133, 2009)(Vermeer et al PLoS One 7:e30447, 2012) (JBC 286: 44976-44987, 2011). Moreover, our data indicate that exosomes (regardless of their EphrinB1 status) activate MAP Kinase signaling in PC12 cells (Figure 6E) suggesting that ligand binding to receptor(s) on PC12 cells is not the only mechanism able to activate signaling cascades important for neuritogenesis.

5) EphrinBs are linked to tumor angiogenesis. Experiments should be performed to exclude that the growth promoting effects of exosomes bearing EphrinB1 in head and neck tumors are linked to increased angiogenesis rather than (or in addition to) innervation.

The reviewer is correct about the link between EphrinBs and angiogenesis and about its contribution to tumor growth. In fact, this is a subaim in our grant and another project in the lab. However, for the revised version of this manuscript, we now limit ourselves to the contribution of exosomes to tumor innervation and make no conclusions on the contribution of innervation to tumor growth. While making this connection is important in cancer biology, it is a topic of its own (one we are currently working on) and that is beyond the scope of this manuscript. As such, all tumor growth curves have been moved to supplemental figures and we make no conclusions about the contribution of tumor innervation to tumor growth.

6) The authors speculate that a binding partner capable of interacting with both full length and deltaECD EphrinB1 could be responsible for neurite outgrowth/innervation induced by the exosomes, rather than EphrinB1 itself. Experiments should address this possibility.

We have performed BioID (a method to screen protein-protein interactions) on EphrinB1 to identify potential binding partners that could mediate neurite outgrowth/innervation. In this method, a fusion protein is generated which consists of the protein of interest (in our case, EphrinB1) fused to BirA, a modified and promiscuous biotin ligase. When expressed in cells, BirA biotinylates proteins that are in close proximity (and presumably interacting with) the protein of interest (Roux et al, JCB 196:801-810, 2012)(Roux Cell Mol Life Sci 70: 3657-3664, 2012). Cells are lysed, biotinylated proteins isolated with streptavidin and identified by Mass spec. We tagged EphrinB1's cytoplasmic domain with BirA and identified potential binding partners. Clustering analysis of our "hits" revealed that many associating proteins function in growth cone activity or neurite outgrowth activity. We present this data below for the reviewers. We are currently validating these "hits" to determine what role they play in exosomal EphrinB1's ability to potentiate neurite outgrowth *in vitro* and tumor innervation *in vivo*. However, these data are not yet mature and thus are not included in this manuscript. As such, we have also removed our speculation about EphrinB1 binding partners and their contribution to tumor innervation.

7) concerns with Figure 4 (now Supplemental Figure 8).

The experiment has been repeated and the new data now replace the previous data. Please see answer to question 4 for Reviewer #1.

8) In Figure 2E (now Figure 6D), the EphrinB1-In antibody labels bands of similar size in cells expressing wild-type EphrinB1 and deltaECD EphrinB1. Shouldn't deltaECD EphrinB1 be smaller in size?

The reviewer is correct. We would like to point out that EphrinB1 is naturally cleaved first by MMPs (extracellular cleavage) and then by the gamma secretase complex (intracellular cleavage). We now show that the EphrinB1-In antibody labels a similar band in deltaECD EphrinB1 as found in naturally processed wildtype EphrinB1. **This change can be found on page 6, lines 256-262.**

9) Did the mEERL tumors with high EphrinB1 or EphrinB1 deltaECD expression (Fig 4, now Supplemental Figure 8) both exhibit faster growth as compared to control tumors, correlating with the similarly increased levels of

beta-III tubulin induced by the two forms of EphrinB1?

This experiment has been repeated. We now include the tumor growth curves from the new experiment (supplemental Figure 8E). These data replicate the original experiment showing that mEERL EphrinB1 and mEERL EphrinB1 Δ ECD tumors grow the fastest of all the mEERL tumors. This is consistent with their increased levels of tumor innervation. However, in this manuscript we make no conclusions about the contribution of tumor innervation to tumor growth.

10) The data in figure 3 (now Figure 7) are disconnected from the manuscript. How is the role of HPV16 in promoting neurite outgrowth connected to EphrinB1?

The connection between HPV16 and EphrinB1 is mediated via HPV16 E6's interaction with the cellular phosphatase, PTPN13. PTPN13 is the phosphatase that regulates EphrinB1. In HPV infected cells, E6 interacts with PTPN13 in a PDZ dependent manner and this interaction results in the degradation of the phosphatase. As a result, EphrinB1 activation becomes dysregulated. **We have added this explanation in the results section page 7, lines 276-289.**

In this manuscript, we investigated the contribution of HPV to tumor innervation simply because we found that exosomes from HPV negative cell lines had significantly lower neurite outgrowth activity than those from HPV positive cell lines; we were intrigued by this finding. The data indicate that the PDZ binding motif of HPV16 E6 is important for neurite outgrowth activity of exosomes and suggests that EphrinB1 activation may be important for neurite outgrowth activity. We are in the process of generating mEERL cell lines in which EphrinB1's tyrosines are changed to either phenylalanine or aspartic acid. Once validated, these cell lines will help define the contribution of activated EphrinB1 to tumor innervation. These studies are on-going in the laboratory and beyond the scope of this manuscript.

11) what are the levels of EphrinB1 in the four head and neck cancer cell lines examined?

We now present western blot analysis of lysates from four human head and neck cancer cell lines and the HTE cells examined (Supplemental Figure 7).

12) Can EphrinB1 over-expression increase the neurite outgrowth promoting activity of exosomes from HPV negative cancer cell lines?

The answer to this important question is yes! We now include this data in figure 7C. We stably over-expressed EphrinB1 in the HPV negative SCC1 cell line (SCC1 EphrinB1). Exosomes harvested from this line and its parental cell line were tested on PC12 cells. Over-expression of EphrinB1 significantly increases neurite outgrowth activity as compared to exosomes from the parental cell line (SCC1) (Figure 7C). We further tested whether this change could affect innervation *in vivo*. Thus, NOD-SCID mice were implanted with 1×10^5 SCC1 parental or SCC1-EphrinB1 cells (N=4 mice/group) and tumor growth monitored for 10 days. The tumor growth curves are now shown in Supplemental Figure 7B and C; SCC1-EphrinB1 tumors grew significantly faster than SCC1 parental tumors. At day 10 post tumor implantation, tumors were harvested, and whole tumor lysate analyzed by western blot for beta-III tubulin, TRPV1, and Tau; signals were normalized to GAPDH and quantified. We now show that SCC1-EphrinB1 tumors are more innervated than SCC1 (parental) tumors (Figures 7D-G). **This change can be found on page 7, lines 300-308 and continuing on page 8, lines 309-**

13) in Fig 1E, exosomes from some patient (Pts 3, 6, 7) tumor tissue do not express detectable EphrinB1 yet they promote neurite outgrowth of PC12 cells. This does not seem consistent with the proposed important role of EphrinB1 in the neurite-inducing effects of patient exosomes.

The data presented in the manuscript demonstrate that **exosomes** promote neurite outgrowth; EphrinB1 is **not** required. Our focus is on exosomes, not EphrinB1. The title of the manuscript indicates this as well (EphrinB1 is not mentioned). Moreover, in figure 6C, we show that exosomes devoid of EphrinB1 (Null1, Null2) still harbor significant neurite outgrowth activity above the negative control (PC12 cells alone). Thus, the fact that the tumor derived exosomes from Pts 3, 6, and 7 lack detectable EphrinB1 in the western blot (Supplemental Figure 1) does not negate our focus but serves, instead, to emphasize it. EphrinB1 is able to induce neurite outgrowth activity but it is not required for it.

14) Additional controls are needed to determine whether the effects of Rab27A and B gene inactivation on tumor growth are due to deficient generation of exosomes or other possible activity of the Rab27 proteins in cancer cells.

Since Rab proteins are involved in many different cellular processes, adequately controlling for each would be difficult. Thus, we have removed an association between exosomes and tumor growth as the manuscript is focused on the contributions of exosomes to innervation (not tumor growth). Moreover, we have included the experiment using GW4869 *in vivo*. In this experiment, exosome release is attenuated without alteration of Rab27 (or any other genes) in mEERL cells, and the result is the same: attenuation of exosome release results in attenuation of tumor innervation (Figure 5G, H). **This change can be found on page 5, lines 203-219 and continuing on page 6, lines 220-222.**

15) It would be interesting to know whether the effects observed are specific for EphrinB1 or whether the related EphrinB2, which is also highly expressed in many cancer cells, is also involved.

We agree with the reviewer that this would be interesting. We have localized EphrinB2 by immunofluorescence in mEERL tumors and found its localization to be different from that of EphrinB1 (data not shown). While not definitive, the data suggest that these two Ephrins might be doing different things in tumor tissue. A study analyzing the contribution of EphrinB2 in exosome mediated tumor innervation, while interesting, is beyond the scope of this manuscript.

Response to reviewers

Reviewer #1

1) In Figure 3, the difference between PC12 and NGF appears significant and was not analyzed. For multiple comparisons, the authors have used two-tailed student's t-test with Benjamini-Hochberg Procedure to adjust p value. This does not seem to take into account multiple groups which are typically analyzed with one-way ANOVA analysis.

The reviewer is correct. We have re-analyzed all the data using one- or two-way ANOVA (as appropriate) with post-hoc Fisher's Least Significant Difference (LSD) test. We report the relevant Fisher's LSD p values in the figure legends. For full transparency, we have now also included supplemental tables (Supplemental Tables 1-8) in which all comparisons for all relevant figures are reported along with the LSD and Benjamini-Hochberg p-values. Figure 6A required two-way ANOVA and was analyzed as such. **These changes can be found in figure legends: Figure 3, pg. 22; Figure 5, page 22; Figure 6, page 23; Figure 7, page 23-24; Figure 8, page 24; Supplemental Figure 2, page 24; Supplemental Figure 8, page 25-26.**

2) In figure 6E, the GAPDH band is not clearly visible. The same for supplemental Figure 1.

We have adjusted the quality of GAPDH in Figure 6E and the bands can now be clearly appreciated. Supplemental Figure 1 does not have GAPDH. The bottom bands are EphrinB1 (not GAPDH) and demonstrate that not all exosome samples are positive for EphrinB1. Those that express EphrinB1 have variable expression (some samples do not express any EphrinB1). We have adjusted the quality of these bands and they can now be clearly appreciated

3) The overall quality of the figures needs to be improved. For example, in Figure 5, the font size of p values and the size of stars is variable.

To improve the quality of all figures, we have removed the p values from all figures and included them, instead, in the figure legends. All stars are now of uniform size. The reviewer was correct, this significantly improved the quality of the figures.

Reviewer #2.

1) However, an important issue raised by the reviewers centered on proving that EphrinB1 played a key role in tumor innervation, as seemed to be suggested in the original manuscript. Now it would appear, based on the new lines of evidence included in the revised manuscript, that EphrinB1 is not playing an absolutely essential role in the ability of exosomes to induce tumor innervation (though it apparently can potentiate the process). Thus, while I would like to give the authors the benefit of the doubt regarding their revised manuscript, considering they have attempted to address a number of the issues raised, it is nonetheless concerning that what seems to be an extremely important question is still unanswered, i.e. the underlying mechanism by which the exosomes stimulate tumor innervation.

We agree with the reviewer. The underlying mechanism by which the exosomes stimulate tumor innervation remains unanswered. This, however, does not minimize the novelty of what we have discovered. A role for tumor released exosomes in the induction of tumor innervation has not been described before. As such, our goal in this initial manuscript focused on validating the observation using a number of different exosome sources (human samples, mouse tumors, mouse cell lines, human cell lines), assays (PC12, NGF neutralization) as well as discrete approaches to block exosome release and test the effects on innervation (Rab27A/B deletion,

GW4869). Our efforts were not limited to *in vitro* assays but also relied heavily on *in vivo* experiments to validate the *in vitro* findings. Moreover, given the central role that exosomes play in this phenomenon coupled with the lack of methodological standardization in the exosome field, we focused on stringently validating the exosomes themselves to ensure that they were, indeed, the vesicles mediating innervation. Our validations included SEM, AFM, nanoparticle tracking analysis, density gradient centrifugation, as well as, western blot analysis. Thus, as with other studies that have uncovered novel biological phenomena, it is beyond the scope of this initial manuscript to also include molecular mechanism. However, we think the reviewer will agree that this does not minimize the novelty of the finding. Instead, this manuscript lays the foundation upon which mechanistic studies will emerge that will define the molecular mechanism by which tumor released exosomes mediate tumor innervation. That is, in fact, where the energy of my laboratory is currently focused.

Reviewer #3.

1) There are still some speculations in the manuscript that shift the focus away from the main message and, in opinion of this reviewer, weaken the manuscript. For example, the authors describe how the effect of EphrinB1 could depend on the E6 HPV viral oncoprotein through the PTPN13 phosphatase (lines 283-289). However, they also show that EphrinB1 confers innervation promoting activity to exosomes derived from HPV-negative tumors (lines 300-304), suggesting that the proposed HPV-PTPN13-EphrinB1 signaling connection may not be critical.

We agree with the reviewer that speculating on the contribution of E6 and its relationship with PTPN13 and subsequently EphrinB1 takes the focus away from the main message of the manuscript. We further agree that it weakens the story. Thus, we have modified the text to focus it back on to the contribution of exosomes (rather than HPV per se) on innervation. The data suggest that E6's PDZBM modulates neurite outgrowth activity of exosomes by modulating exosome content or quality (e.g. phosphorylation status of exosomal proteins). While this may occur via E6's effects on PTPN13 and subsequently EphrinB1, we have removed this speculation from the text. **This change can be found on page 7, line 287-289.** To further strengthen the manuscript, we have also removed text in the introduction and the interpretation sections regarding HPV16 E6, PTPN13 and EphrinB1. We believe these changes simplify the text as well as the data and keep the main point of the manuscript in focus.

2) The discussion on neuro-immune interactions could be shortened.

The discussion on neuro-immune interactions has been shortened as suggested (**page 9, lines 387-390**).

3) Analysis of exosomes derived from only two tumors is not sufficient to conclude that exosomes derived from HPV-negative tumors have impaired ability to promote innervation. Indeed, it appears that if HPV-negative tumors express EphrinB1, their exosomes have good ability to promote innervation. The text should be modified accordingly.

The reviewer makes a very valid point. We have removed the text speculating that HPV-negative tumors have impaired ability to promote innervation. We have also pointed out the possibility that HPV-negative tumors that harbor elevated EphrinB1 may, in fact, be potentiated in tumor innervation (as our SCC1-EphrinB1 data suggest). These changes can be found on **page 7, lines 291-293 and lines 301-304; page 8, lines 317-318.**

4) The difference could be prolonged, rather than short-term (5 min), MAP kinase pathway activation, since PC12 differentiation/neurite outgrowth requires prolonged MAP kinase pathway activation.

The reviewer makes a very good point. The literature is clear that sustained MAP kinase signaling is key for inducing differentiation of PC12 cells. Our experiment (Figure 6E) shows that EphrinB1 is not required for the initiation of MAP kinase signaling in PC12 cells (since EphrinB1 null exosomes similarly induce P-Erk1/2 activation). It remains possible, however, that EphrinB1 allows for prolongation of this signaling that could then result in enhanced differentiation of PC12 cells. We have added this possibility in the text along with relevant references (**page 7, lines 272-274**).

5) Is the type of innervation in the HPV-negative SCC1-EphrinB1 tumors different from that observed in the HPV-positive tumors, since the levels of the TRPV1 sensory marker are not increased concomitant with the increased in the beta-III tubulin and Tau innervation markers?

The reviewer makes a very interesting point. It is possible that the type of innervation is changed in the HPV-negative SCC1-*EphrinB1* tumors. We have currently have no reason to believe that this does or does not occur in vivo but have wondered that ourselves. We have considered the question that if sensory innervation is blocked by chemical or physical means, would exosomes recruit different types of nerves to the tumor? If tumor innervation is indeed found to be critical for tumor growth, the prediction would be that the tumor would evolve mechanisms (possibly by modulating exosome cargo) to recruit different nerves. These are certainly questions of interest that we are currently pursuing.

6) Supplemental figure 8A,D shows that the level of TRPV1 are lower in the deltaECD tumors compared to EphrinB1 tumors. Suppl. Fig. 8A also shows much lower levels of Tau in deltaECD compared to EphrinB1 tumors, although the quantification in Suppl. Fig. 8C does not indicate a significant difference. Thus, it is not clear that EphrinB1 deltaECD promotes innervation in vivo as strongly as full-length EphrinB1.

The reviewer makes a very good observation. Quantification of neuronal markers (β -III tubulin and tau) indicates that mEERL EphrinB1 and mEERL deltaECD tumors are equally innervated (Suppl Fig. 8B and C). Please keep in mind that the western blot shown in Supplemental Figure 8A is only one of five that were quantified. Thus, while it may appear as if the Tau in deltaECD is lower in the western blot presented (Supplemental Figure 8A), the quantification is based on five different western blots since n=5 tumors/group were analyzed. This has been more clearly noted in the figure legend (**page 25, lines 1096-1097**). It is clear, however, that deltaECD tumors have significantly less TRPV1 compared to EphrinB1 tumors (Suppl Fig. 8D). These data suggest that while the *extent* of innervation may not be different between mEERL EphrinB1 and mEERL deltaECD tumors, the *type* of innervation may be different in the deltaECD tumors. This is a similar point the reviewer raised about the SCC1-*EphrinB1* tumors (see question 5). Moreover, while not the focus of this manuscript, the correlation we see between the extent of tumor innervation and tumor growth remains when one looks at the tumor growth curves for mEERL EphrinB1 and mEERL deltaECD (Suppl Fig 8E). Again, a switch in the type of innervation is something of great interest to the lab and an area we are actively pursuing. Moreover, our quantification of beta-III tubulin and TRPV1 co-stained fibers indicates that not all “twigs” are TRPV1 positive (line 103). While 83.7% of beta-III tubulin fibers are TRPV1 positive, 16.3% of beta-III tubulin positive fibers were TRPV1 negative. Thus, it is not unreasonable to hypothesize that alterations in exosomal signals (e.g. full-length vs deltaECD) could alter the % of TRPV1 negative fibers in the tumor. The main point of this manuscript (that tumor released exosomes induce tumor innervation) remains unchanged but the observation opens up very interesting questions. We have modified the text to make this more clear (**page 8, lines 331-335**).

7) “axonogenesis” already means “generation of new axons”, thus the prefix “neo” is not necessary.

We have changed all “neo-axonogenesis” to “axonogenesis.” **These changes can be found throughout the manuscript.**

8) It should be mentioned that EphrinB1, like other ephrins, is also frequently associated with growth cone collapse and inhibition of axon outgrowth.

We have added text about EphrinB1 and other ephrins in growth cone collapse and inhibition of axonal outgrowth (**page 2, lines 66-67**).

9) Is it of concern that some beta-III tubulin and TRPV1 immunoreactivities appear to associate with “nucleated fibers”, suggesting that not only neurites express the two markers?

We were initially surprised to see nucleated cells co-labeling for beta-III tubulin and TRPV1 as we felt these markers would only identify cells of neuronal origin and label axons in the tumor. Our use of an additional neuronal marker (tau) for quantifying tumor innervation becomes all the more important to ensure the neuronal nature of the innervation we are studying. Whether the nucleated cells are cancer associated fibroblasts (CAF) remains unclear. However, the standard markers for CAFs do not include beta-III tubulin or TRPV1. CAF markers include: alpha-smooth muscle actin (SMA), fibroblast activation protein (FAP), tenascin-C, desmin, PDGF-R, fibroblast specific protein 1 and other markers (Shiga, K et al *Cancers* 7: 2443-2458, 2015)(Huang L et al *World Journal of Gastroenterology* 20: 17804-17818, 2014)(Paulsson, J and Micke P, *Seminars in Cancer Biology* 25: 61-68, 2014). The markers typically used to define CAFs are alpha-SMA and FAP. Given this, we have removed the speculation on the identity of the nucleated cells but retained our observation of their presence. This change can be found on **page 3, line 92**.

10) It is difficult to understand what is shown in some of the panels in Suppl Figs. 5 and 6 based on the explanations in the figure legends. For Suppl Fig. 5C, is this perhaps due to mislabeling of the lanes?

We apologize for the confusion. While Suppl Fig 5C was not incorrectly labeled, the + and – were inadvertently shifted making the figure very difficult to understand. We have fixed this spacing and now the figure is correct. We have also restructured Supplemental Figures 5 and 6 to make them easier to understand. The sequence data was somewhat redundant with the electropherogram and, thus, confusing. Therefore, we eliminated it leaving only the electropherogram. We also added a schematic of the EphrinB1 exons to make it easier to see where the changes in the CRISPR clones were generated. We believe these figures are now easier to understand.

Point-by-point response to reviewer

1) It is not clear why the authors conclude that EphrinB1 lacking the extracellular domain can potentiate the neurite-promoting activity of HNSCC-derived exosomes when their data seem to support the opposite conclusion. First, exosomes derived from mEERL EphrinB1 Δ ECD cells do not promote PC12 neurite outgrowth compared to exosomes derived from parental cells (Fig. 6d). Second, expression of β III-tubulin, tau and TRPV1 (used as innervation markers) is not significantly different in tumors from mEERL EphrinB1 Δ ECD cells and from parental mEERL cells (Suppl. Fig. 8 b,c,d). Third, only a very small amount of the truncated EphrinB1 seems to be present in the exosomes from Δ ECD cells (Fig. 6d). It is also difficult to envision how exosomal EphrinB1 lacking the ectodomain could induce innervation. Despite these caveats, the authors state that “the extent of innervation was similar in mEERL EphrinB1 and EphrinB1 Δ ECD tumors” but “the type of innervation may be different” (page 8, second paragraph); that “While the PC12 assay would have predicted that mEERL EphrinB1 Δ ECD tumors to be sparsely innervated in vivo, we instead found that they were as innervated.....”; and that “full length EphrinB1 is not required for tumor innervation” (page 9, second paragraph).

When analyzing the in vivo data (Supplemental Figure 8), we first focused on the extent of innervation, that is, the amount of β -III tubulin and tau present in whole tumor lysates. For simplicity, we will only be concerned with mEERL parental, mEERL EphrinB1 and mEERL EphrinB1 Δ ECD tumors. When one looks at Supplemental Figure 8b, one finds that there is a significant increase in β -III tubulin expression in the mEERL EphrinB1 tumors as compared to the mEERL parental tumors. However, there is no significant difference between β -III tubulin expression between mEERL EphrinB1 tumors and mEERL EphrinB1 Δ ECD tumors; in other words, mEERL EphrinB1 and mEERL EphrinB1 Δ ECD tumors express equal amounts of β -III tubulin. That is why we state that the extent of innervation was similar in mEERL EphrinB1 and mEERL EphrinB1 Δ ECD tumors (as β -III tubulin is a general marker for the nerves). The same holds true when one looks at the quantification for Tau (Supplemental Figure 8c). Thus, these two pieces of data support each other and indicate that mEERL EphrinB1 and mEERL EphrinB1 Δ ECD tumors are equally innervated. When one looks Supplementary Figure 8d, the focus is now on the specific type of innervation (sensory, TRPV1). Here, we see that while mEERL EphrinB1 tumors have a significantly increased TRPV1 expression, mEERL parental and mEERL EphrinB1 Δ ECD tumors do not. In fact, mEERL parental and mEERL EphrinB1 Δ ECD tumors are not significantly different from each other. Taken together, the data indicate that mEERL EphrinB1 tumors are more innervated (β -III tubulin, Tau) and that this increased innervation is more sensory (TRPV1) in nature than the other tumors. The data also indicate that while mEERL EphrinB1 Δ ECD tumors are just as highly innervated as mEERL EphrinB1 tumors (β -III tubulin, Tau), this higher level of innervation is NOT equally high in TRPV1 expression. That is to say, that there are nerves in mEERL EphrinB1 Δ ECD tumors that are β -III tubulin positive, Tau positive but TRPV1 negative. These may be sympathetic, parasympathetic, or TRPV1 negative sensory in nature, though this was not tested. Thus, we concluded that the type of innervation may be different in mEERL EphrinB1 Δ ECD tumors as compared to mEERL EphrinB1 tumors. Moreover, while not the focus of this manuscript, the correlation we see between the extent of tumor innervation and tumor growth remains when one looks at the tumor growth curves for mEERL EphrinB1 and mEERL Δ ECD (Suppl Fig 8E). These data are consistent with equal innervation but a possible change in the type of innervation. We have added text to the manuscript to make this point more clear.

These *in vivo* data indicate that an elevated tumor innervation capacity (i.e. the ability to induce increased β -III tubulin and Tau expression) do not require full length EphrinB1. This is supported by the PC12 data (Figure 6C) where exosomes from mEERL EphrinB1 Δ ECD induce neurite outgrowth similar to those of mEERL parental. mEERL parental exosomes contain basal levels of full length EphrinB1 while mEERL EphrinB1 Δ ECD exosomes possess extracellularly truncated EphrinB1 (with no full length EphrinB1) yet both retain the capacity to induce equal amounts of neurite outgrowth from PC12 cells. Thus, these data also indicate that full length EphrinB1 is not required for neurite outgrowth activity. The amount of neurite outgrowth induced by EphrinB1 Δ ECD exosomes is, however, less than what is seen *in vivo*. So, as stated in the manuscript, the *in vitro* PC12 assay does not predict the *in vivo* tumor innervation data. As noted above, Supplemental Figures 8B and C indicate that mEERL EphrinB1 Δ ECD tumors are not significantly different from mEERL EphrinB1 tumors with respect to β -III tubulin and Tau expression. We emphasize again that the PC12 assay is just an assay and while it provides information on neurite outgrowth capacity of exosomes, it lacks the microenvironment that exosomes encounter *in vivo* and which can modulate the tumor innervation that is seen.

There are a number of discrepancies between the full western blots provided with this version of the manuscript and the relative figures.

2) First, the full blots for the control plasma exosomes do not look the same as those shown in Fig. 2d (for CD9, even the relative intensity of the band in the two lanes is different).

The reviewer is correct. We inadvertently chose the wrong exposure of the full western blots. This has now been corrected.

3) Second, in the full TRPV1 blot (mis-labeled Tau) relative to Fig. 5B, how do the authors know that the band they chose is a specific band?

We apologize for the mislabeling. This has now been corrected. We include below (and in the full westerns PDF) a western blot containing whole (mouse) brain lysate which is blotted for TRPV1 showing its size (approx. 100kD). In addition, knock-out of TRPV1 demonstrates loss of the 100kD band in a publication by Chen et al (Cadiovascular Diabetology 14: 22-35 (2015)).

4) Third, in the NGF blot relative to Fig. 6B, how do the authors know which among the many bands represents NGF? Could it be that the NGF band is the one slightly above the one indicated, which is present also in the exosomes? A lane containing purified NGF should be run next to the sample lanes to determine the size of NGF. Or a second (better?) antibody should be used for blotting (the anti-NGF antibody used for this blot is not specified in the Methods).

The reviewer is correct. The band we had marked as NGF is incorrect and has now been fixed. An elegant study published in 1996 by Seidah et al (Biochem J 314: 951-960) demonstrates that NGF is generated as a precursor, pro-NGF, which runs at a molecular weight of 35kD. This precursor is glycosylated, generating a 42.5kD intermediate which is then processed to 13.5 and 16.5kD forms. The full western for Figure 6B shows two bands running below the 17kD marker corresponding to the two processed forms of pro-NGF which are present in the mEERL parental and mEERL EphrinB1 whole cell lysate but absent in their exosomes. We have added a line of text to make this more explicit and have also included the reference mentioned above. The full western blot is included here:

While we agree with the reviewer that the western blot has many bands, we did try several anti-NGF antibodies as follows: Abcam ab6199, ThermoFisher PA1-18378, Abcam ab49205. These antibodies demonstrated similar results on western blot.

5) Fourth, the identity of the samples is designated differently in the CD9 and CD81 full western blots relative to the blots shown in Fig. 6D.

We thank the reviewer for catching this mistake. The two full westerns were inadvertently flipped. We have fixed this problem.

6) Fifth, the TRPV1 full western blot for Fig. 7C raises the possibility that the TRPV1 antibody used is not suitable for western blotting. How do the authors know which of the many similarly labeled bands is TRPV1?

In the full westerns for Figure 7C, we have now included an additional western blot in which we included whole (mouse) brain lysate and blotted with the same anti-TRPV1 antibody. This blot clearly shows that TRPV1 runs at approximately 100Kd. In addition, we also include a reference in the full westerns to a publication by Chen et al (Cardiovascular Diabetology 14: 22-35, 2015), in which they show the loss of the 100Kd band in tissue from TRPV1 knock-out mice.

7) Sixth, for the full Tau blot relative to Fig. 7C, the authors chose a band of ~ 80 kD (indicated by the arrow), but the molecular weight of Tau isoforms is reported to range from 45 to 65 kD (perhaps corresponding to some of the lower molecular weight bands also labeled in the blot).

The reviewer is correct. While the quantification was performed on the correct blot, the wrong Tau band was presented in the figure. We have now fixed this problem and thank the reviewer for pointing it out. We also include in the Full westerns for Figure 7C, an additional western blot in which whole (mouse) brain lysate was run and blotted with the same Tau antibody. This blot clearly shows that Tau runs at approximately 55kD (as noted by the reviewer).

8) The authors should check carefully the blots presented in the supplementary figures for similar problems.

We have gone carefully over all figures and believe they are now all correct and accurate.

9) Supplementary Table 1 shows the statistical significance of comparisons between samples using two methods. Non-significant p values obtained with one of the two methods (Benjamini-Hochberg), however, are not considered in the analysis of the results or to reach conclusions. Why are these p values not considered? If there is a reason not to consider them, then why are they provided in the table?

The Benjamini-Hochberg corrected p-values were included because a reviewer had asked for them in a previous revised submission. We agree with the reviewer, however that they are not necessary. In fact, the Fisher's LSD test was chosen for its statistical power over other more conservative post hoc comparisons (like Benjamini-Hochberg). In light of this comment, we have removed the Benjamini-Hochberg p values from the Supplementary tables, leaving only the LSD p values.

10) Abstract, line 3. "disease" should be "tumors".

The change has been made.

11) Abstract, line 3. "induce" should be "can induce". Likely not all tumors release exosomes that promote tumor innervation (see, for example, the HPV negative tumors). In addition, not all tumors are innervated.

The text in line three reads "We hypothesize that nerves are acquired by a tumor-induced process, called axonogenesis." Making the change from "induce" to "can induce" makes the text awkward. However, we agree with the reviewer and changed the sentence to: "We hypothesize that in some tumors, nerves are acquired by a tumor-induced process, called axonogenesis." We hope the reviewer agrees that this change still is consistent with his comment.\

12) Abstract, lines 12 and 13. As written, this sentence states that exosomal cargo only potentiates tumor innervation, implying that exosomes induce tumor innervation through a mechanism that is independent of their cargo. This is probably not what the authors mean.

We agree with the reviewer. We have changed the sentence into two sentences as follows "These findings indicate that tumor released exosomes induce tumor innervation. Exosomes containing EphrinB1 potentiate this activity."

13) Abstract line 13. “potentiate” should be “potentiates”.

This sentence has been changed. Please see response to point 11.